# Functional Renormalisation Group approach to shell models of turbulence

Côme Fontaine[1], Malo Tarpin[2], Freddy Bouchet[2], and Léonie Canet[1,3]⋆

**1** Univ. Grenoble Alpes, CNRS, LPMMC, 38000 Grenoble, France
**2** ENS Lyon, 69000 Lyon, France
**3** Institut Universitaire de France, 75000 Paris, France
⋆ leonie.canet@lpmmc.cnrs.fr

## Abstract

Shell models are simplified models of hydrodynamic turbulence, retaining only some essential features of the original equations, such as the non-linearity, symmetries and quadratic invariants. Yet, they were shown to reproduce the most salient properties of developed turbulence, in particular universal statistics and multi-scaling. We set up the functional renormalisation group (FRG) formalism to study generic shell models. In particular, we formulate an inverse RG flow, which consists in integrating out fluctuation modes from the large scales (small wavenumbers) to the small scales (large wavenumbers), which is physically grounded and has long been advocated in the context of turbulence. Focusing on the Sabra shell model, we study the effect of both a large-scale forcing, and a power-law forcing exerted at all scales. We show that these two types of forcing yield different fixed points, and thus correspond to distinct universality classes, characterised by different scaling exponents. We find that the power-law forcing leads to dimensional (K41-like) scaling, while the large-scale forcing entails anomalous scaling.

# 1 Introduction and summary of results

Hydrodynamic turbulence is a multi-scale non-linear phenomenon which still lacks a complete theoretical understanding. One of the salient stumbling blocks is the calculation of intermittency effects [1]. An homogeneous and isotropic three-dimensional (3D) turbulent flow exhibits at large Reynolds number (Re) universal properties for scales in the inertial range, that is away from the large scale $L$ where the forcing is exerted and from the Kolmogorov scale $\eta$ where dissipation occurs. In this range, empirical evidence suggests that the structure functions $S_p(r)$ of velocity increments behave as power-laws

$$S_p(r) = \left\langle \left| \boldsymbol{v}(t, \boldsymbol{x} + \boldsymbol{r}) - \boldsymbol{v}(t, \boldsymbol{x}) \right|^p \right\rangle \sim r^{\zeta_p}, \quad r = |\boldsymbol{r}|, \tag{1}$$

with universal exponents $\zeta_p$ which are independent of the forcing mechanism. Kolmogorov established in his celebrated 1941 theory (K41) [2,3] the exact result $\zeta_3 = 1$ in the limit of infinite Re. Then assuming self-similarity in the statistical sense, he inferred $\zeta_p = p/3$. However, the statistics of the velocity in real turbulent flows do not obey K41 theory. Accumulated evidence from experiments and numerical simulations shows that the structure functions exhibit anomalous scaling with exponents $\zeta_p \neq p/3$, which reveals the multi-scale or multi-fractal nature of developed turbulence [1]. The deviations from K41 theory are referred to as intermittency effects, and their calculation from Navier-Stokes equation remains an unsolved problem.

The calculation of anomalous exponents requires in general elaborated methods, such as the Renormalisation Group (RG), conceived by Wilson [4], which has been pivotal to understand the emergence of universality and anomalous critical exponents at an equilibrium second-order phase transition. Shortly after the work of Wilson, the analogy between the phenomenology of equilibrium critical phenomena and fully developed turbulence – both featuring scale invariance, universality and anomalous scaling – was pointed out, and motivated the use of RG techniques for turbulence [5–7]. Let us emphasise an essential physical difference between the two. In equilibrium critical phenomena, the scale invariance with anomalous exponents emerges at large distances $\ell \gg a$, where $a$ is typically the lattice spacing, and the large-scale statistical properties are universal with respect to the microscopic (UV) details. In

turbulence, the anomalous behaviour is observed at small scales $\ell \ll L$, where $L$ is the integral scale, and the statistics are universal with respect to the precise form of the (IR) forcing mechanism. The standard Wilson's RG for equilibrium critical phenomena consists in averaging over small-scale (UV) fluctuation modes to build up the effective theory for the large-scale (IR) modes. Correspondingly, it was suggested that the most appropriate RG for turbulence should be an inverse flow, which averages over large-scale fluctuations to construct the effective theory for the small-scale modes [8, 9]. Although physically appealing, this idea has only been concretely implemented for the Kraichnan model for passively advected scalar, for which direct RG is also successful [10, 11].

In fact, disregarding this subtlelty, intensive efforts were devoted to applying standard RG techniques to study turbulence [5–7]. It turns out that these efforts have been largely thwarted by the absence of a small parameter to control perturbative expansions [12–14]. The usual strategy to circumvent this issue in perturbative implementation of the RG for three-dimensional hydrodynamic turbulence has been to artificially introduce an expansion parameter $\varepsilon$ via a long-range self-similar forcing with a power-law spectrum $N(k) \sim k^{4-d-\varepsilon}$ [6, 7, 12–15]. However, the physical case of a large-scale forcing corresponds to the uncontrolled limit $\varepsilon \to 4$. This raises the question of whether the actual short-range fixed point, corresponding to a large-scale forcing, can be captured by such an expansion. This issue has been investigated in 3D numerical simulations [16, 17], and revisited in a shell model [18], which showed that when the self-similar forcing dominates, the statistics are Gaussian and the scaling exponents $\zeta_p$ are dimensional (at least for low orders $p$) whereas for a large-scale forcing the statistics are intermittent and the exponents are anomalous. Conversely, numerical simulations in 2D turbulence found that the power-law dimensional spectrum predicted by the perturbative RG is never observed neither in the direct nor in the inverse cascade [19, 20].

Since the work of Wilson, the formulation of the RG has deeply evolved, and matured into a very versatile and powerful framework, which allows for functional and non-perturbative implementations of the RG procedure (we refer to [21–24] for general reviews on FRG). In turbulence, the FRG has yielded two important results. The first one is to show the existence of the RG fixed point which describes developed turbulence for a forcing concentrated at large scale, which is the relevant case for physical situations [25–27]. This fixed point yields the exact four-fifth law and allows for the accurate computation of the Kolmogorov constant [28]. The second and prominent result derived from FRG is the expression of the general spatio-temporal dependence of multi-point multi-time Eulerian correlations [29]. This expression is exact in the limit of large wavenumbers, and provides not only the general form of the correlations, but also the associated constants. These predictions were quantitatively confirmed in direct numerical simulations for the two-point and three-point correlation functions [30, 31], and also for passive scalar turbulence [32, 33]. Yet intermittency exponents, which are associated with structure functions, *i.e. equal-time properties*, have not been computed yet within the FRG framework, and constitute an important goal which we address in this work.

Rather than directly tackling the Navier-Stokes equation, a fruitful alternative approach to gain insights in the statistical properties of turbulence is to conceive simplified models, which involve a considerably reduced number of degrees of freedom. Among such models are the one-dimensional Burgers equation [34], the Constantin-Lax-Majda model [35], or shell models [36,37]. Shell models, originally introduced by Gledzer [38], Desnyansky and Novikov [39] are inspired by the energy cascade picture, and are formulated as an Euler-like dynamics of a few Fourier modes $v(t, k_n)$ of the velocity field whose wavenumbers are logarithmically spaced: $k_n = k_0 \lambda^n$ with $\lambda > 1$. For simplicity, $v$ and $k_n$ are simply taken as scalars rather than three-dimensional vectors, such that the geometry is lost, and they do not correspond to any exact projection of the Navier-Stokes equation [40–42]. Yet, they preserve the features of the dynamical equations which are intuitively thought as important, such as the structure of

the quadratic non-linearity, symmetries, and quadratic inviscid invariants. Indeed, they were shown to generate multi-scaling, and for some values of the parameters they even quantitatively reproduce the anomalous exponents measured in hydrodynamic turbulence [43–45]. This was numerically demonstrated in particular for the Gledzer-Yamada-Ohkitani (GOY) shell model [38, 43], and its improved version proposed by L'vov *et al* called the Sabra model [45]. The latter allows for efficient and more accurate numerical determination of the anomalous exponents by suppressing some oscillatory behaviour of the original GOY model.

One can also derive analytical results on intermittency for some specific models, such as the spherical shell model in the large $N$ limit [46, 47], or shell models with a Hamiltonian structure [48] or specifically designed solvable models [49]. Note that one can also rigoroulsy construct shell models which preserve the exact form of the original equation of motions [42]. However, for the GOY and Sabra models whose properties most closely resemble the ones of hydrodynamic turbulence, there is no analytical result on their anomalous scaling exponents. Moreover, despite some early attempts [50], the existence of a fixed point describing their universal properties has been only postulated, but never demonstrated.

In this paper, we employ FRG methods to study shell models, focusing on the Sabra shell model, with the aim of computing its structure functions. We exploit the versatility of the FRG framework to investigate two aspects in particular, the role of the forcing profile – large scale *vs* power-law, and the influence of the direction of the RG flow – direct *vs* inverse. Concerning the forcing, we study the turbulence generated by both a forcing concentrated on large scales, termed short-range (SR), and a self-similar forcing characterised by an algebraic spectrum of the form $N(k_n) \sim k_n^{-\rho}$, termed long-range (LR). Regarding the direction of the RG flow, besides the study of the direct RG (from UV to IR), which is the conventional one, we also formulate the inverse RG (from IR to UV), which had never been implemented before in the FRG framework to the best of our knowledge.

Let us summarise our main results. The first one is that our analysis shows that the LR and SR forcings yield two distinct fixed points: the LR fixed point has normal dimensional scaling, whereas the SR one is anomalous. An important consequence is that they do not coincide even in the limit where $\rho \to 0$ (which is equivalent to $\varepsilon \to 4$ for 3D turbulence). It follows that anomalous intermittency exponents cannot be computed from taking the limit $\rho \to 0$ of the LR fixed-point. This suggests that this could also be the case for hydrodynamic turbulence.

The second result is that we find that the inverse RG is better suited to study the SR fixed point. We explicitly compute the universal properties of the turbulent state at this fixed point, in particular the structure functions of order $p = 2, 3, 4$, within an approximation which consists in a vertex expansion. We show that this fixed point indeed yields anomalous scaling, *i.e.* the exponents $\zeta_p$ differ from K41 scaling, although at this level of approximation, they are uni-fractal, akin in a $\beta$-model [51], rather than fully multi-fractal. We discuss possible improvements of the approximation which could lead to multi-fractality.

The rest of the paper is organised as follows. The Sabra shell model and its key properties are reviewed in Sec. 2. We then use the path integral formalism to derive the field theory for shell models. In Sec. 3, we introduce the functional renormalisation group and expound its formulation for shell models, both in a direct and in an inverse flow. We present in Sec. 4 the approximation studied in this work, which consists in a vertex expansion, and derive the corresponding FRG flow equations. We describe in Sec. 5 our results for the direct RG flow. We first recover the standard perturbative results for a LR forcing, and then show that the presence of a SR forcing yields a crossover to a different fixed point. We explain why this fixed point is more conveniently accessed within a inverse flow, which we implement in Sec. 6. We compute the associated structure functions and show that they exhibit anomalous exponents, although not multi-scaling. Technical details are reported in the Appendices.

## 2 Sabra shell model and field theory

### 2.1 Sabra shell model

The Sabra shell model describes the dynamics of a set of complex amplitudes $(v_n(t))_{n\in\mathbb{Z}}$, which represent Fourier modes of the velocity in shells of wavenumbers $k_n = k_0 \lambda^n$ with $\lambda > 1$. The time evolution of these amplitudes is given by a set of non-linear coupled ordinary differential equations [45]:

$$
\begin{cases}
\partial_t v_n = B_n[v, v^*] - \nu k_n^2 v_n + f_n \\
B_n[v, v^*] = i\left[ a k_{n+1} v_{n+2} v_{n+1}^* + b k_n v_{n+1} v_{n-1}^* - c k_{n-1} v_{n-1} v_{n-2} \right],
\end{cases}
\tag{2}
$$

where $\nu$ is the viscosity and $f_n$ a forcing to maintain a turbulent stationary state. The three parameters $a, b, c$ must satisfy the constraint $a + b + c = 0$ to ensure that the inviscid ($\nu = 0$) and unforced ($f = 0$) version of (2) formally conserves two quadratic invariants. These are the total energy and total "helicity" (in analogy with 3D hydrodynamic turbulence), defined respectively as

$$
E = \sum_n v_n v_n^*, \qquad H = \sum_n \left(\frac{a}{c}\right)^n v_n v_n^*.
\tag{3}
$$

Since the equations can be rescaled to set one of the parameters to unity, and another one is fixed by the conservation constraint, we conform to the conventional choice of setting $a = 1$ and $b = -(1 + c)$. Note that the $k_n$ are positive and finite, but one may also define this shell model on a wavenumber space containing the zero mode.

The model (2) possesses four symmetries, which are the invariance under time translations, space translations, time dilatations and discrete space dilatations. Since $(v_n(t))_{n\in\mathbb{Z}}$ represents a sequence of Fourier modes, the translations in space correspond to phase shifts of the form $v_n \longrightarrow v_n' = e^{i\theta_n} v_n$. The Sabra shell model is invariant under these shifts provided that the phases $(\theta_n)_{n\in\mathbb{Z}}$ satisfy the recurrence relation $\theta_{n+2} = \theta_{n+1} + \theta_n$. The solutions of this equation are of the form

$$
\theta_n = \frac{1}{\sqrt{5}}\left[ \theta_0(\varphi_+ \varphi_-^n - \varphi_- \varphi_+^n) + \theta_1(\varphi_+^n - \varphi_-^n) \right],
\tag{4}
$$

where $\varphi_\pm = (1 \pm \sqrt{5})/2$ are the solutions of $\varphi^2 = 1 + \varphi$. As shown in Ref. [45], the independence of the field configurations with respect to $\theta_0$, $\theta_1$ implies that they must satisfy a rule of conservation of a quasi-momentum $\kappa_n = \pm\varphi^n$, which is that "the sum of incoming quasi-momenta (associated with a $v$) is equal to the sum of outgoing quasi-momenta (associated with a $v^*$)", as stated in [45]. This was identified as a key advantage of the Sabra shell model compared to the GOY one. Indeed, it is natural to assume that the statistics of the model respect this symmetry. The conservation of the quasi-momentum then entails strict constraints on the non-vanishing field configurations, contrarily to the law of conservation of momentum for the Fourier transform in the continuum space.

The analogy with Fourier modes should not be pushed too far nonetheless. Indeed, in shell models, the shell variables $v_n$ are rather surrogates for the velocity increments in 3D hydrodynamical turbulence and as such their moments are the analogues of the structure functions. For $p = 2$ and $p = 3$, the only structure functions compatible with the conservation of the quasi-momentum are:

$$
S_2(k_n) = \left\langle v_n v_n^* \right\rangle
\tag{5}
$$

$$
S_3(k_n) = \text{Im}\left\langle v_{n+1}^* v_n v_{n-1} \right\rangle.
\tag{6}
$$

For higher orders $p$, one defines the following structure functions:

$$
S_p(k_n) =
\begin{cases}
\langle |v_n|^p \rangle & \text{if } p \text{ is even} \\
\text{Im}\langle v_{n+1}^* v_n v_{n-1} |v_n|^{p-3} \rangle & \text{if } p \text{ is odd}.
\end{cases}
\tag{7}
$$

The great interest of the GOY and Sabra shell models is that these structure functions exhibit a very similar behaviour as in hydrodynamic turbulence. At high Reynolds number (large scale forcing and small viscosity $\nu$), they display an inertial range of shell indices with universal scaling laws characterised by anomalous exponents [36, 37, 45]. For the Sabra model, all structure functions behave as:

$$S_p(k_n) = C_p \left( \frac{\varepsilon}{k_n} \right)^{\frac{p}{3}} \left( \frac{k_0}{k_n} \right)^{\delta \zeta_p} , \tag{8}$$

where $C_p$ are dimensionless constants. We define the anomalous exponents as $\zeta_p = \frac{p}{3} + \delta \zeta_p$. The equivalent of Kolmogorov K41 theory yields for the shell models $\zeta_p = \frac{p}{3}$, and thus $\delta \zeta_p$ is the deviation from Kolmogorov scaling. The equivalent of the exact 4/5th law [3] also holds in these models, and one can show that the third order structure function is given by the exact expression

$$S_3(k_n) = \frac{1}{2k_n(a-c)} \left( -\varepsilon + \left( \frac{a}{c} \right)^n \delta \right) , \tag{9}$$

where $\varepsilon$ is the mean energy dissipation rate and $\delta$ is the mean helicity dissipation rate [45]. One can tune the large scale forcing in order to cancel this last contribution, or alternatively one can decrease its impact by choosing $\left| \frac{a}{c} \right| < 1$ [45]. Apart from $\zeta_3 = 1$ which is fixed by (9), the other exponents are observed to be anomalous, *i.e.* $\zeta_p \neq \frac{p}{3}$. While the K41 dimensional analysis and Eq. (9) hold for any values of the parameters (provided $a + b + c = 0$), the values of the anomalous exponents depend on the choice of parameters $c$ and $\lambda$ [37]. They are quantitatively very close to the ones observed in fluid turbulence for the specific choice $c = -0.5$ and $\lambda = 2$ [45]. All our computations are carried out using this choice of parameters.

For fixed parameters, the behaviour in the inertial range is universal. This means that the exponents $\zeta_p$ do not depend on the viscosity $\nu$ and on the precise form of the forcing $f_n$ as long as it only acts on large scales. While one usually introduces a deterministic forcing on the first two shells tuned to eliminate the spurious helicity flux, we rather consider, as common in field theoretical approaches of turbulence, a stochastic forcing, which can be chosen as Gaussian without loss of generality [50]. Its covariance is defined as

$$\left\langle f_n(t) f_{n'}(t') \right\rangle = 2N(k_n)\delta(t - t')\delta_{nn'} . \tag{10}$$

For a large scale forcing, $N(k_n)$ is concentrated at the integral scale, *i.e.* it is non-zero for shells with $k_n \simeq L^{-1}$. In practice, we use the profile

$$N_{L,n} = D_{\text{SR}} h(Lk_n) , \qquad h(x) = x^2 e^{-\alpha_1 (x - \alpha_2)^2} , \tag{11}$$

where $D_{\text{SR}}$ is the forcing amplitude, $\alpha_1$ and $\alpha_2$ are dimensionless coefficients. In all our calculation, we use $\alpha_1 = 10^{-2}$ and $\alpha_2 = 4$. For a power-law forcing acting on all scales, $N(k_n)$ is defined as

$$N_{\rho,n} = D_{\text{LR}} \left( \frac{k_n}{k_0} \right)^{-\rho} . \tag{12}$$

## 2.2 Path integral formalism

The Sabra shell model (2) in the presence of the stochastic forcing (10) takes the form of a set of coupled Langevin equations. These stochastic equations can be cast into a path integral formalism using the Martin-Siggia-Rose-Janssen-de Dominicis procedure [52–54], which was implemented in the context of shell models in Ref. [46, 50]. This procedure relies on the introduction of response fields $\bar{v}_n$ and $\bar{v}_n^*$ which play the role of Lagrange multipliers to enforce the equations of motion. Collecting the different fields in a multiplet $V = (v, v^*, \bar{v}, \bar{v}^*)$ to

alleviate notation, the generating functional associated with the dynamics of the shell model is given by

$$\mathcal{Z}[J] = \int \mathcal{D}[V] e^{-\mathcal{S}[V] + \sum_{n,\alpha} \int_t V_{\alpha,n} J_{\alpha,n}}, \tag{13}$$

where we introduced the source multiplet $J = (j, j^*, \bar{j}, \bar{j}^*)$, and Greek indices $\alpha = \{1, 2, 3, 4\}$ span the fields in the multiplet, whereas $n \in \mathbb{Z}$ refers to the shell index. The action for the Sabra shell model is given by

$$\mathcal{S}[V] = \sum_n \int_t \left\{ \left[ \bar{v}_n^* \left( \partial_t v_n + \nu k_n^2 v_n - B_n[v, v^*] \right) - \frac{1}{2} N(k_n) \bar{v}_n \bar{v}_n^* \right] + \text{c.c.} \right\}, \tag{14}$$

where c.c. denotes the complex conjugate. This action is general for shell models, each model differing only in the precise form of the non-linear interaction $B[v, v^*]$. We also consider the generating functional $\mathcal{W} = \ln \mathcal{Z}$. In the framework of equilibrium statistical mechanics, the functional $\mathcal{Z}$ is the partition function of the system, and $\mathcal{W}$ its free energy. In probability theory, $\mathcal{Z}$ is analogous to the characteristic function, *i.e.* the generating function of moments, while $\mathcal{W}$ is the generating function of cumulants. They are simply promoted to functionals in the context of field theory since the random variables are here fluctuating spacetime-dependent fields. The central interest of these functionals is that all the statistical properties of the system can be derived from them. In particular the $m$-point correlation functions (generalisation of moments), or $m$-point connected correlation functions (generalisation of cumulants), can be computed by taking $m$ functional derivatives of $\mathcal{Z}$, respectively $\mathcal{W}$, with respect to the sources, and evaluating them at zero sources.

The presence of forcing and dissipation ensures that the system reaches a non-equilibrium statistically stationary state. In the following, we thus assume that the statistics are invariant under time translations, and we introduce the Fourier transform of the fields in time. We use the following conventions:

$$f(\omega) \equiv \mathcal{F}[f(t)] = \int_t e^{i\omega t} f(t) = \int_{\mathbb{R}} dt \, e^{i\omega t} f(t)$$

$$f(t) \equiv \mathcal{F}^{-1}[f(\omega)] = \int_\omega e^{-i\omega t} f(\omega) = \int_{\mathbb{R}} \frac{d\omega}{2\pi} e^{-i\omega t} f(\omega). \tag{15}$$

The Sabra field theory can be studied by means of RG techniques, which we introduce in the following section.

## 3 Functional renormalisation group formalism

The functional renormalisation group is a modern formulation of the Wilsonian renormalisation group, which allows for functional and non-perturbative approximations [21–24]. The general idea underlying the renormalisation group is to perform a progressive averaging of the fluctuations of a system, may they be thermal, quantum or stochastic, in order to build the effective theory of the system at the macroscopic scale. The RG procedure is endowed through FRG with a very versatile framework, which is used in a wide range of applications, both in high-energy physics (quantum gravity and QCD), condensed matter, quantum many-body systems and statistical mechanics, including disordered and non-equilibrium problems (for reviews see [21–24]).

In the context of turbulence, the FRG was employed to study homogeneous and isotropic turbulence in several works [25–27, 29, 30, 32, 55, 56]. As mentioned in the introduction, the

main achievements so far are twofold: to show the existence of the fixed point describing the universal properties of turbulence forced at large scales; to provide the expression of the spatio-temporal multi-point multi-time Eulerian correlations, which is exact in the limit of large wavenumbers [28]. In this work, we resort to the FRG method to study shell models of turbulence, focusing on equal-time properties, *i.e.* structure functions.

## 3.1 Direct and inverse RG for shell models

Wilson's RG procedure consists in achieving *progressively* the integration of the fluctuations in the path integral (13). In the conventional, or direct, RG, this integration starts from the large wavenumber (UV) modes and includes more and more smaller (IR) ones. In order to achieve this progressive integration, one introduces a continuous wavenumber scale parameter $\kappa$, and a cut-off term which suppresses all fluctuation modes on shells $k_n$ below the RG scale $\kappa$. This cut-off term, also called regulator, takes the form of an additional quadratic term $\Delta S_\kappa$ (analogous to a mass) in the path integral action, which reads for the shell model

$$\Delta S_\kappa[V] = \frac{1}{2} \sum_n \int_t V_{\alpha,n} \mathcal{R}_{\kappa,n}^{\alpha\beta} V_{\beta,n}, \tag{16}$$

where the regulator matrix, with the fields ordered as $(v, v^*, \bar{v}, \bar{v}^*)$, reads:

$$\mathcal{R}_{\kappa,n} = \begin{pmatrix} 0 & 0 & 0 & R_{\kappa,n} \\ 0 & 0 & R_{\kappa,n} & 0 \\ 0 & R_{\kappa,n} & 0 & 0 \\ R_{\kappa,n} & 0 & 0 & 0 \end{pmatrix}. \tag{17}$$

We emphasise that the $vv^*$ sector must vanish for causality reasons [57], while any contribution involving two conjugated field or two non-conjugated fields is incompatible with the conservation of quasi-momentum. One also could consider a $\bar{v}\bar{v}^*$ cut-off, but it is not necessary so we choose not to include it for simplicity. The $v\bar{v}^*$ term can be interpreted as a scale-dependent damping friction term at scale $\kappa$ [27]. Note that we restrict to memoryless cut-off functions, that is cut-off functions which are white in time. This choice simplifies the calculations and was shown to yield reliable results in many applications of FRG to non-equilibrium systems [24].

In the standard direct RG, the cut-off function $R_{\kappa,n}$ is required to be very large for modes with $k_n \lesssim \kappa$ such that the contribution of these modes in the path integral is suppressed, and to vanish for modes with $k_n \gtrsim \kappa$ such that the contribution of these modes is unaltered. Thus, at a given RG scale $\kappa$, only the fluctuations on shells $n \gtrsim \ln(\kappa)/\ln(\lambda)$ are integrated in the path integral, and as $\kappa \to 0$, all fluctuations are progressively included. A common choice which fulfils these requirements is the Wetterich cut-off

$$R_{\kappa,n} = \nu k_n^2 \, r\left(\frac{k_n}{\kappa}\right), \quad r(x) = \frac{\alpha}{e^{x^2} - 1}, \tag{18}$$

where $\alpha$ is a free parameter.

In order to formulate an inverse RG flow, one would like to carry out the progressive integration of the fluctuation modes in the reverse direction, that is from the IR wavenumbers (large scales) to the UV ones (small scales). Thus, one needs to choose a cut-off function which suppresses the contributions of modes $k_n \gtrsim \kappa$ while it does not affect modes with $k_n \lesssim \kappa$. We propose as a suitable choice the following function:

$$R_{\kappa,n} = \nu \frac{k_0^4}{\kappa^2} \, r\left(\frac{k_n}{\kappa}\right), \quad r(x) = \alpha \tanh(\beta x^2), \tag{19}$$

where $\alpha$ and $\beta$ are free parameters. The pre-factor is chosen to be homogeneous to viscosity×wave-number[2] as required by dimensional arguments. [1]

## 3.2 Exact FRG flow equations

In both the direct and the inverse RG flow, the addition of $\Delta S_\kappa$ turns $\mathcal{Z}$ into a scale-dependent generating functional:

$$Z_\kappa[J] = \int \mathcal{D}[V]\, e^{-\mathcal{S}[V] - \Delta \mathcal{S}_\kappa[V] + \sum_{n,\alpha} \int_t V_{\alpha,n} J_{\alpha,n}}\,. \tag{20}$$

In the FRG formalism, the central object is the effective average action $\Gamma_\kappa$, defined as the (modified) Legendre transform of $\mathcal{W}_\kappa = \ln \mathcal{Z}_\kappa$ as:

$$\Gamma_\kappa[U] = \left[ \sum_{n,\alpha} \int_t U_{\alpha,n} J_{\alpha,n} \right] - \mathcal{W}_\kappa[J] - \Delta S_\kappa[U], \tag{21}$$

where the value of $J$ is fixed by the relation

$$U \equiv \langle V \rangle = \frac{\delta \mathcal{W}_k[J]}{\delta J}, \tag{22}$$

such that $U$ corresponds to the expectation value of $V$. The functional $\Gamma_\kappa$ turns out to be well-suited to implement functional and non-perturbative approximation schemes [21, 24]. The effective average action is solution of the exact Wetterich equation:

$$\partial_\kappa \Gamma_\kappa = \frac{1}{2} \mathrm{Tr} \sum_n \int_t \partial_\kappa \mathcal{R}_{\kappa,n}(t) \cdot G_{\kappa,n}(t), \qquad G_{\kappa,n} = \left[ \Gamma^{(2)}_{\kappa,n} + \mathcal{R}_{\kappa,n} \right]^{-1}, \tag{23}$$

where $\Gamma^{(2)}_\kappa$ is the Hessian of the effective average action. This flow equation allows one to continuously interpolate between the microscopic theory, which is here the shell model action (14) and the full effective theory we aim at when all fluctuations have been integrated out. To ensure these limits, the cut-off function has to satisfy additional requirements. For the direct flow, $R_{\kappa,n}$ should vanish when $\kappa \to 0$ (or $\kappa = \Lambda_{\mathrm{IR}}$ some very small wavenumber) such the regulator is removed and one obtains the full effective action $\Gamma_{\Lambda_{\mathrm{IR}}} \equiv \Gamma$, while $R_{\kappa,n}$ should diverge (or be very large) for $\kappa \to \infty$ (or $\kappa = \Lambda_{\mathrm{UV}}$ some very large wavenumber) such that all the fluctuations are frozen and one recovers the microscopic action $\Gamma_{\Lambda_{\mathrm{UV}}} \equiv \mathcal{S}$ [24]. For an inverse flow, the cut-off should satisfy reverse limits, *i.e.* $R_{\kappa,n} \to \infty$ (or is very large) when $\kappa = \Lambda_{\mathrm{IR}}$ such that $\Gamma_{\kappa = \Lambda_{\mathrm{IR}}} \equiv \mathcal{S}$ while $R_{\kappa,n} \to 0$ when $\kappa \to \Lambda_{\mathrm{UV}}$ such that $\Gamma_{\Lambda_{\mathrm{UV}}} \equiv \Gamma$. One can check that the cut-off functions defined in (18) and (19) satisfy these requirements.

The functional $\Gamma$ is called in field theory the generating functional of one-particle irreducible correlation functions. It corresponds in the context of equilibrium statistical mechanics to the Gibbs free energy. In non-equilibrium systems, it still incorporates the effects of fluctuations at all scales, and allows one to compute all the statistical properties of the system. Since the functionals $\Gamma_\kappa$ and $\mathcal{W}_\kappa$ are related by a Legendre transform (21), the knowledge of the set of $\Gamma^{(m)}_\kappa$ is equivalent to the knowledge of the set of $\mathcal{W}^{(m)}_\kappa$. In particular, the structure functions can be computed from the propagator $G_\kappa$ and the vertices $\Gamma^{(m)}_\kappa$, see Sec. 3.4. The flow equation for the $\Gamma^{(m)}_\kappa$ can be obtained by taking $m$ functional derivatives of (23) with

---

[1] Let us make a technical comment anticipating on the exact flow equation (23). This equation involves the scale derivative $\partial_\kappa \mathcal{R}_{\kappa,n}$ of the regulator. For the choice (19), this derivative does not vanish at large wavenumber, but rather tends to a constant. While this is not a problem for shell models, it could be detrimental in a continuous model for the convergence of the integration over wavevectors. In this case, an alternative cut-off function should be chosen.

respect to the fields. For instance, this yields for the flow equation of the two-point function $\Gamma_\kappa^{(2)}$ the following flow equation

$$\partial_\kappa \tilde{\Gamma}_{\kappa,nn'}^{U_\alpha U_{\alpha'}}(\omega) = \mathrm{Tr} \sum_{m_i} \int_{\omega_1} \partial_\kappa \tilde{\mathcal{R}}_{\kappa,m_1 m_2}(\omega_1) \cdot \tilde{G}_{\kappa,m_2 m_3}(\omega_1) \cdot \left[ \frac{1}{2} \tilde{\Gamma}_{\kappa,nn'm_3 m_4}^{U_\alpha U_{\alpha'},(2)}(\omega,-\omega,\omega_1) \cdot \tilde{G}_{\kappa,m_4 m_1}(\omega_1) \right.$$

$$\left. - \tilde{\Gamma}_{\kappa,nm_3 m_4}^{U_\alpha,(2)}(\omega,\omega_1) \cdot \tilde{G}_{\kappa,m_4 m_5}(\omega+\omega_1) \cdot \tilde{\Gamma}_{\kappa,n'm_5 m_6}^{U_{\alpha'},(2)}(-\omega,\omega+\omega_1) \cdot \tilde{G}_{\kappa,m_6 m_1}(\omega_1) \right],$$

$$(24)$$

where the tilded quantities are defined as

$$\Gamma_{\kappa,n_1\cdots n_n}^{(m)}(\omega_1,\cdots,\omega_m) = 2\pi\delta(\omega_1+\cdots+\omega_m)\tilde{\Gamma}_{\kappa,n_1\cdots n_m}^{(m)}(\omega_1,\cdots,\omega_{m-1}). \tag{25}$$

They depend on one less frequency than their non-tilded counterpart owing to frequency conservation. We recall that we are explicitly focusing on the stationary state. The vertices $\tilde{\Gamma}_\kappa^{(3)}$ and $\tilde{\Gamma}_\kappa^{(4)}$ are here put in the form of $4 \times 4$ matrices with one, e.g. $\tilde{\Gamma}_\kappa^{U_\alpha,(2)}$, respectively two, e.g. $\tilde{\Gamma}_\kappa^{U_\alpha U_{\alpha'},(2)}$, legs fixed to the external fields $U_\alpha$ and $U_{\alpha'}$ and external indices $n$ and $n'$. The dot thus represents a matrix product in this notation. This flow equation can also be represented using Feynman diagrams as:

$$\partial_\kappa \Gamma_{\kappa,nn'}^{U_\alpha U_{\alpha'}} = \qquad\qquad - \frac{1}{2} \qquad\qquad , \tag{26}$$

where the vertices are given by the $\Gamma_\kappa^{(m)}$, the internal lines correspond to the propagator $G_\kappa$, and the cross stands for the derivative of the regulator $\partial_\kappa \mathcal{R}_\kappa$. Since the propagator enters in all the flow equations, we first determine its general structure.

## 3.3 General form of the propagator

The propagator $G_\kappa$ is defined as the inverse of the Hessian of $\Gamma_\kappa + \Delta\mathcal{S}_\kappa$. From general considerations of causality, and translational invariance in space, there are only two independent terms in the Hessian of the effective average action. Using translational invariance in time, they take the following form in Fourier space

$$\mathcal{F}\left[\frac{\delta^2 \Gamma_\kappa}{\delta\bar{u}_n(t)\bar{u}_{n'}^*(t')}\right](\omega,\omega') = \bar{\gamma}_{\kappa,n}(\omega)\delta_{nn'}2\pi\delta(\omega+\omega')$$

$$\mathcal{F}\left[\frac{\delta^2 \Gamma_\kappa}{\delta u_n(t)\bar{u}_{n'}^*(t')}\right](\omega,\omega') = \gamma_{\kappa,n}(\omega)\delta_{nn'}2\pi\delta(\omega+\omega'), \tag{27}$$

where $\mathcal{F}$ denotes the Fourier transform defined in (15). Thus, the Hessian of $\Gamma_\kappa$ is given by

$$\tilde{\Gamma}_{\kappa,nn'}^{(2)}(\omega) = \delta_{nn'} \begin{pmatrix} 0 & 0 & 0 & \gamma_{\kappa,n}(\omega) \\ 0 & 0 & \gamma_{\kappa,n}(\omega) & 0 \\ 0 & \gamma_{\kappa,n}(-\omega) & 0 & \bar{\gamma}_{\kappa,n}(\omega) \\ \gamma_{\kappa,n}(-\omega) & 0 & \bar{\gamma}_{\kappa,n}(-\omega) & 0 \end{pmatrix}. \tag{28}$$

Then, as $\tilde{\Gamma}_\kappa^{(2)} + \tilde{\mathcal{R}}_\kappa$ is diagonal in shell index, its inverse is simply a matrix inverse, given by

$$\tilde{G}_{\kappa,nn'}(\omega) = \delta_{nn'} \begin{pmatrix} 0 & \bar{g}_{\kappa,n}(\omega) & 0 & g_{\kappa,n}(\omega) \\ \bar{g}_{\kappa,n}(-\omega) & 0 & g_{\kappa,n}(\omega) & 0 \\ 0 & g_{\kappa,n}(-\omega) & 0 & 0 \\ g_{\kappa,n}(-\omega) & 0 & 0 & 0 \end{pmatrix}. \tag{29}$$

with

$$g_{\kappa,n}(\omega) = \frac{1}{\gamma_{\kappa,n}(-\omega) + R_{\kappa,n}}, \qquad \bar{g}_{\kappa,n}(\omega) = -\frac{\bar{\gamma}_{\kappa,n}(-\omega)}{\left|\gamma_{\kappa,n}(\omega) + R_{\kappa,n}\right|^2}. \tag{30}$$

## 3.4 Expression of the structure functions

The statistical properties of the model can be computed at the end of the flow, in the limit $\kappa \to \Lambda_{\mathrm{IR}}$ for the direct flow or $\kappa \to \Lambda_{\mathrm{UV}}$ for the inverse one, that is once all the fluctuations have been averaged out. In particular, if the RG flow reaches a fixed point (which is expected from the observation of universality and scale invariance), the structure functions can be calculated from the fixed-point functions. The structure functions $S_p(k_n)$ are equal-time correlation functions. As mentioned in Sec. 3.2, all the correlation functions can be calculated from the set of vertex functions $\Gamma^{(m)}$. The relations between the $S_p$ and the $\Gamma^{(m)}$ can be simply deduced by taking functional derivatives of the Legendre transform (21) relating their respective generating functionals $\mathcal{W}_\kappa$ and $\Gamma_\kappa$. One has the following chain-rule relation

$$\frac{\delta \mathcal{W}_\kappa}{\delta J_{\alpha,n}(t)} = -\int_{t'} \sum_{n',\alpha'} \frac{\delta U_{\alpha',n'}(t')}{\delta J_{\alpha,n}(t)} \frac{\delta \Gamma_\kappa}{\delta U_{\alpha',n'}(t')} = -\int_{t'} \sum_{n'} G_{\kappa,nn'}^{U_\alpha U_{\alpha'}}(t,t') \Gamma_{\kappa,n'}^{U_{\alpha'}}(t'), \tag{31}$$

neglecting the term linear in $J$ in (21) which plays no role for higher-order derivatives, as well as the $\Delta \mathcal{S}_\kappa$ term since it vanishes at the end of the flow. The second order structure function $S_2(k_n)$ is given by

$$S_2(k_n) = \left\langle v_n(t) v_n^*(t) \right\rangle = \int_\omega \frac{\delta^2 \mathcal{W}_\kappa}{\delta j_n \delta j_n^*} \bigg|_{\kappa \to \kappa_{\mathrm{final}}} = \int_\omega \bar{g}_{\kappa,n}(\omega) \bigg|_{\kappa \to \kappa_{\mathrm{final}}}, \tag{32}$$

where $\kappa_{\mathrm{final}}$ denotes the final RG scale, either $\Lambda_{\mathrm{UV}}$ or $\Lambda_{\mathrm{IR}}$.

Using the relation (31), one obtains for the third-order structure function

$$S_3(k_n) = \mathfrak{Im}\left\langle v_{n-1} v_n v_{n+1}^* \right\rangle = \mathfrak{Im}\left[ \frac{\delta^3 \mathcal{W}_\kappa}{\delta j_{n-1}(t) j_n(t) j_{n+1}^*(t)} \right] \bigg|_{\kappa \to \kappa_{\mathrm{final}}}$$

$$= -\mathfrak{Im} \sum_{m_i} \sum_{\alpha,\beta,\gamma} \int_{\omega_1,\omega_2} \tilde{G}_{\kappa,n-1 m_1}^{u U_\alpha}(\omega_1) \tilde{G}_{\kappa,n m_2}^{u U_\beta}(\omega_2) \tilde{\Gamma}_{\kappa,m_1 m_2 m_3}^{U_\alpha U_\beta U_\gamma}(\omega_1,\omega_2) \tilde{G}_{\kappa,n+1 m_3}^{u^* U_\gamma}(-\omega_1-\omega_2) \bigg|_{\kappa \to \kappa_{\mathrm{final}}}. \tag{33}$$

Similarly, a general fourth-order structure function is defined as

$$\left\langle U_{\alpha_1,n_1}(t) U_{\alpha_2,n_2}(t) U_{\alpha_3,n_3}(t) U_{\alpha_4,n_4}(t) \right\rangle = \left[ \frac{\delta^4 \mathcal{W}_\kappa}{\delta J_{\alpha_1,n_1}(t) J_{\alpha_2,n_2}(t) J_{\alpha_3,n_3}(t) J_{\alpha_4,n_4}(t)} \right] =$$

$$-\sum_{m_i} \sum_{\beta_i} \int_{\omega_i} \tilde{G}_{\kappa,n_1 m_1}^{U_{\alpha_1} U_{\beta_1}}(\omega_1) \tilde{G}_{\kappa,n_2 m_2}^{U_{\alpha_2} U_{\beta_2}}(\omega_2) \tilde{G}_{\kappa,n_3 m_3}^{U_{\alpha_3} V_{\beta_3}}(\omega_3) \tilde{\Gamma}_{\kappa,m_1 m_2 m_3 m_4}^{U_{\beta_1} U_{\beta_2} U_{\beta_3} U_{\beta_4}}(\omega_1,\omega_2,\omega_3) \tilde{G}_{\kappa,n_4 m_4}^{U_{\alpha_4} U_{\beta_4}}(\omega_4)$$

$$+\sum_{m_i} \sum_{\beta_i} \int_{\omega_i} \tilde{G}_{\kappa,n_1 m_1}^{U_{\alpha_1} U_{\beta_1}}(\omega_1) \tilde{G}_{\kappa,n_2 m_2}^{U_{\alpha_2} U_{\beta_2}}(\omega_2) \tilde{\Gamma}_{\kappa,m_1 m_2 m_5}^{U_{\beta_1} U_{\beta_2} U_{\beta_5}}(\omega_1,\omega_2) \tilde{G}_{\kappa,m_5 m_6}^{U_{\beta_5} U_{\beta_6}}(\omega_1+\omega_2)$$

$$\times \tilde{\Gamma}_{\kappa,m_6 m_3 m_4}^{U_{\beta_6} U_{\beta_3} U_{\beta_4}}(\omega_1+\omega_2,\omega_3) \tilde{G}_{\kappa,n_3 m_3}^{U_{\alpha_3} U_{\beta_3}}(\omega_3) \tilde{G}_{\kappa,n_4 m_4}^{U_{\alpha_4} U_{\beta_4}}(\omega_4) \quad + \text{permutations}, \tag{34}$$

where we have defined $\omega_4 \equiv -\omega_1 - \omega_2 - \omega_3$. If there are no 4-point vertices in a given approximation, only the terms in the third and fourth lines of (34) contribute. With only these terms, the purely local $S_4$ defined as

$$S_4(k_n) = \left\langle v_n v_n v_n^* v_n^* \right\rangle \tag{35}$$

vanishes because of constraints from spatial translational invariance for the $\tilde{\Gamma}^{(3)}$. In this case, we consider another configuration permitted by quasi-momentum conservation (see App. A) defined as

$$S_4(k_n) = \left\langle v_n v_{n+1} v_{n+1}^* v_{n+1}^* \right\rangle. \tag{36}$$

When 4-point vertices are included in a given approximation, both definitions (35) and (36) entail a non-vanishing contribution from (34). We checked that they always exhibit the same scaling exponent.

## 4 Approximation scheme within the FRG

The flow equation (23) is exact, but it is a functional non-linear partial differential equation which obviously cannot be solved exactly. In particular, as usual with non-linear equations, it is not closed, *i.e.* the flow equation for a given $\Gamma_\kappa^{(m)}$ depends on higher-order vertex functions $\Gamma_\kappa^{(m+1)}$ and $\Gamma_\kappa^{(m+2)}$ (as manifest in (24) for $m = 2$), such that it leads to an infinite hierarchy of flow equations. Thus one has to resort to some approximations. The key asset of the FRG formalism is that (23) lends itself to functional and non-perturbative approximation schemes. The most common one is called the derivative expansion as it consists in expanding $\Gamma_\kappa$ in powers of spatial and temporal derivatives, while retaining the functional dependence in the fields. However, in the context of turbulence, we expect a rich behaviour at small scales, *i.e.* large wavenumbers, which is not *a priori* accessible through such a scheme. The alternative common approximation scheme rather consists in a vertex expansion, *i.e.* an expansion of $\Gamma_\kappa$ in powers of the fields, while retaining a functional dependence in the derivatives, or equivalently in wavenumbers and frequencies [21, 24].

In fact, for Navier-Stokes turbulence, it was shown that, owing to the general properties of the regulator and to the existence of extended symmetries of the Navier-Stokes field theory, the flow equation for any $\Gamma_\kappa^{(m)}$ can be closed (*i.e.* it does not depend any longer on higher-order $\ell > m$ vertices) for large wavenumbers without further approximation than taking the large wavenumber limit [28, 29]. Its solution at the fixed-point yields the exact general expression at leading order in large wavenumber for the spatio-temporal dependence of arbitrary $m$-point Eulerian correlation functions. This expression gives in particular the rigorous form of effects associated with random sweeping. However, this leading order term vanishes for equal times. It means that it does not encompass equal-time correlation functions, in particular structure functions. The random sweeping effect thus hides the intermittency corrections, in the sense that the latter are contained in sub-leading terms compared to sweeping. As a consequence, anomalous exponents have not been computed yet in the FRG framework for turbulence.

In shell models, the effect of sweeping is concentrated in the zero mode, and because of the locality of the interactions in shell indices, it does not affect the other modes. Hence, the intermittency effects are *a priori* more directly accessible, in the sense that they are not contained in subleading terms compared to the sweeping because the latter is absent. The purpose of this work is to assess this hypothesis and access intermittency effects in these simplified models.

### 4.1 Vertex expansion

We focus in the following on equal-time quantities, namely the structure functions, and compute them within the vertex expansion scheme. Since we aim at determining the wavenumber dependence of structure functions, we keep the most general wavenumber dependence of the vertices, while we only retain their bare (microscopic) frequency dependence. This assumption seems reasonable, but it would be interesting to improve on it in future work. Furthermore, we truncate the expansion at order two in each field, *i.e.* we neglect vertices with more than two

velocity fields or more than two response fields, which we refer to as "bi-quadratic" approximation. We emphasise that in all previous FRG studies of 3D turbulence, $\Gamma_\kappa$ was truncated at quadratic order in the fields [25–27]. In these works, the fixed point was shown to exhibit K41 scaling, without anomalous corrections. Here in contrast, we include vertices up to quartic order, albeit restricting to power two of velocity and response velocity fields. This level of truncation is already quite involved, although of course in principle it would be desirable to study the effect of higher-order vertices. We show that accounting for vertices beyond the bare one indeed yield an anomalous scaling for $S_2$, which is already an important step.

The vertex expansion at this bi-quadratic order can be expressed in the form of the following ansatz for the effective average action $\Gamma_\kappa$:

$$
\Gamma_\kappa = \Gamma_{2,\kappa} + \Gamma_{3,\kappa}^{uu\bar{u}} + \Gamma_{3,\kappa}^{u\bar{u}\bar{u}} + \Gamma_{4,\kappa}^{uu\bar{u}\bar{u}}
$$

$$
\Gamma_{2,\kappa} = \sum_n \int_t \left\{ \bar{u}_n^* \left[ f_{\kappa,n}^\lambda \partial_t u_n + f_{\kappa,n}^\nu u_n - \frac{1}{2} \bar{u}_n^* f_{\kappa,n}^D \bar{u}_n \right] + \text{c.c.} \right\}
$$

$$
\Gamma_{3,\kappa}^{uu\bar{u}} = \sum_n \int_t \left\{ -i\bar{u}_n^* \left[ \chi_{\kappa,n+1}^a k_{n+1} u_{n+2} u_{n+1}^* + \chi_{\kappa,n}^b k_n u_{n+1} u_{n-1}^* - \chi_{\kappa,n-1}^c k_{n-1} u_{n-1} u_{n-2} \right] + \text{c.c.} \right\}
$$

$$
\Gamma_{3,\kappa}^{u\bar{u}\bar{u}} = \sum_n \int_t \left\{ -i u_n^* \left[ \gamma_{\kappa,n+1}^a \bar{u}_{n+2} \bar{u}_{n+1}^* + \gamma_{\kappa,n}^b \bar{u}_{n+1} \bar{u}_{n-1}^* - \gamma_{\kappa,n-1}^c \bar{u}_{n-1} \bar{u}_{n-2} \right] + \text{c.c.} \right\}
$$

$$
\Gamma_{4,\kappa}^{uu\bar{u}\bar{u}} = \sum_n \int_t \left\{ \bar{u}_n^* \left[ \mu_{\kappa,n}^a \bar{u}_n u_n u_n^* + \mu_{\kappa,n}^b \bar{u}_n^* u_n u_n \right] + \text{c.c.} \right\} \tag{37}
$$

The term $\Gamma_{2,\kappa}$ is the most general quadratic term compatible with causality and quasi-momentum conservation, keeping the frequency dependence as in the bare action. It involves three renormalisation functions $f_{\kappa,n}^\nu, f_{\kappa,n}^D, f_{\kappa,n}^\lambda$ which retain a full functional dependence in the wavenumber $k_n$. Their initial condition at the beginning of the flow is

$$
f_{\kappa_{\text{init}},n}^\nu = \nu k_n^2, \qquad f_{\kappa_{\text{init}},n}^\lambda = 1, \qquad f_{\kappa_{\text{init}},n}^D = D_{\text{SR}} h(k_n/k_0) + D_{\text{LR}} \left( \frac{k_n}{k_0} \right)^{-\rho}, \tag{38}
$$

where $\kappa_{\text{init}} = \Lambda_{\text{UV}}$ in the direct RG flow and $\kappa_{\text{init}} = \Lambda_{\text{IR}}$ in the inverse RG flow. The condition initial for $f_\kappa^D$ includes both a SR forcing with amplitude $D_{\text{SR}}$ and a LR algebraic one with amplitude $D_{\text{LR}}$. Each can be studied separately by setting the amplitude of the other to zero.

Similarly, the cubic terms present in the microscopic action with coupling $a$, $b$, and $c$ are promoted in $\Gamma_{3,\kappa}^{uu\bar{u}}$ to full functions of the wavenumber through the renormalisation process. Their initial condition is thus

$$
\chi_{\kappa_{\text{init}},n}^a = a, \qquad \chi_{\kappa_{\text{init}},n}^b = b, \qquad \chi_{\kappa_{\text{init}},n}^c = c. \tag{39}
$$

The other term $\Gamma_{3,\kappa}^{u\bar{u}\bar{u}}$ includes all other possible cubic vertices (apart from a three-response-field vertex) which are not present in the microscopic action but which are generated by the RG flow. The initial condition of the corresponding renormalisation functions $\gamma_{\kappa,n}^a, \gamma_{\kappa,n}^b, \gamma_{\kappa,n}^c$, is thus zero. For all the terms contained in $\Gamma_{2,\kappa}$, $\Gamma_{3,\kappa}^{uu\bar{u}}$ and $\Gamma_{3,\kappa}^{u\bar{u}\bar{u}}$, the only approximation is the simplification of the frequency sector. Indeed, in principles, higher order time derivatives could be included, that is a frequency dependence of the renormalisation functions.

For the quartic vertices, we restrict to those with at most two velocities or response velocities (bi-quadratic order). The configurations permitted by the conservation of quasi-momentum are established and listed in App. A. This shows that there are many possible 4-point vertices. The only ones which can give explicit non-vanishing contributions to the fourth-order structure function (35) are the purely local ones $(n, n, n, n)$ or the non-local ones of the form $(n_1, n_1, n_2, n_2)$. The latter would be associated with renormalisation functions depending on

two wavenumbers $(n_1, n_2)$, which increases the complexity of the numerical resolution of the flow. As a first approximation, we restrict ourselves in this work to the purely local contribution, which are the ones contained in $\Gamma_{4,\kappa}^{uu\bar{u}\bar{u}}$ with renormalisation functions $\mu_{\kappa,n}^a$ and $\mu_{\kappa,n}^b$. As we discuss in the following, the non-local contributions may be necessary to obtain a full multi-fractal behaviour. This is left for future explorations.

## 4.2 Derivation of the flow equations

The flow equations of the renormalisation functions can be calculated from the exact equation (23). The elements entering the right-hand side of this equation are the propagator and the vertices. The propagator is defined by (29). Taking two functional derivatives of (37) with respect to $\bar{u}_n^*, u_{n'}$, and $\bar{u}_n^*, \bar{u}_{n'}$, one obtains

$$\gamma_{\kappa,n}(\omega) = i\omega f_{\kappa,n}^\lambda + f_{\kappa,n}^\nu, \qquad \bar{\gamma}_{\kappa,n}(\omega) = -f_{\kappa,n}^D, \tag{40}$$

with $\gamma_\kappa$ and $\bar{\gamma}_\kappa$ defined in (27).

To obtain the expression of the vertices, one takes additional functional derivatives of (37) with respect to the appropriate fields. For instance,

$$\Gamma_{\kappa,n_1 n_2 n_3}^{uu\bar{u}^*}(\omega_1, \omega_2, \omega_3) \equiv \mathcal{F}\left[\frac{\delta^{(3)}\Gamma_\kappa}{\delta u_{n_1}(t_1)\delta u_{n_2}(t_2)\delta \bar{u}_{n_3}^*(t_3)}\right]$$
$$= i\chi_{n_3}^c k_{n_3}\left(\delta_{n_3 n_1+1}\delta_{n_3 n_2+2} + \delta_{n_3 n_1+2}\delta_{n_3 n_2+1}\right)2\pi\delta(\omega_1 + \omega_2 + \omega_3)$$

$$\Gamma_{\kappa,n_1 n_2 n_3}^{uu^*\bar{u}^*}(\omega_1, \omega_2, \omega_3) \equiv \mathcal{F}\left[\frac{\delta^{(3)}\Gamma_\kappa}{\delta u_{n_1}(t_1)\delta u_{n_2}^*(t_2)\delta \bar{u}_{n_3}^*(t_3)}\right]$$
$$= -i\left(\chi_{n_3+1}^a k_{n_3+1}\delta_{n_3 n_2-1}\delta_{n_3 n_1-2} + \chi_{n_3}^b k_{n_3}\delta_{n_3 n_2+1}\delta_{n_3 n_1-1}\right)2\pi\delta(\omega_1 + \omega_2 + \omega_3). \tag{41}$$

The other vertices can be calculated in a similar way by taking the appropriate functional derivatives of the ansatz (37). One obtains for the 3-point vertices:

$$\tilde{\Gamma}_{\kappa,n_1 n_2 n_3}^{u^*\bar{u}\bar{u}^*} = -i\gamma_{n_1}^a \delta_{n_1 n_3-1}\delta_{n_1 n_2-2} - i\gamma_{n_1}^b \delta_{n_1 n_3+1}\delta_{n_1 n_2-1}$$
$$\tilde{\Gamma}_{\kappa,n_1 n_2 n_3}^{u^*\bar{u}\bar{u}} = i\gamma_{n_1}^c\left(\delta_{n_1 n_3+1}\delta_{n_1 n_2+2} + \delta_{n_1 n_3+2}\delta_{n_1 n_2+1}\right). \tag{42}$$

and for the 4-point vertices:

$$\tilde{\Gamma}_{\kappa,n_1 n_2 n_3 n_4}^{uu^*\bar{u}\bar{u}^*} = \mu_{n_1}^a \delta_{n_2 n_1}\delta_{n_3 n_1}\delta_{n_4 n_1}, \qquad \tilde{\Gamma}_{\kappa,n_1 n_2 n_3 n_4}^{uu\bar{u}^*\bar{u}^*} = \mu_{n_1}^b \delta_{n_2 n_1}\delta_{n_3 n_1}\delta_{n_4 n_1}. \tag{43}$$

From (40), one deduces that the flow equations of the renormalisation functions of the quadratic part can be obtained from the flow of the two-point functions $\Gamma_\kappa^{u\bar{u}^*}$ and $\Gamma_\kappa^{\bar{u}\bar{u}^*}$ as

$$\partial_\kappa f_{\kappa,n}^\nu = \mathfrak{Re}\left[\partial_\kappa \tilde{\Gamma}_{\kappa,nn}^{u\bar{u}^*}\right] \qquad \partial_\kappa f_{\kappa,n}^\lambda = \frac{1}{i\omega}\mathfrak{Im}\left[\partial_\kappa \tilde{\Gamma}_{\kappa,nn}^{u\bar{u}^*}\right] \qquad \partial_\kappa f_{\kappa,n}^D = -\partial_\kappa \tilde{\Gamma}_{\kappa,nn}^{\bar{u}\bar{u}^*}. \tag{44}$$

The exact flow equations for $\Gamma_\kappa^{u\bar{u}^*}$ and $\Gamma_\kappa^{\bar{u}\bar{u}^*}$ are given by (24), which can be projected onto the ansatz (37) by inserting into the flow equation (24) the explicit expressions for the propagator $G_\kappa$ given by (29), (30) and (40) and for the vertices $\Gamma_\kappa^{(3)}$ and $\Gamma_\kappa^{(4)}$ given by (41), (42) and (43). The expression of these flow equations is given in App. B.

The flow equations for the other renormalisation functions can be obtained in a similar way, by taking higher functional derivatives of the exact flow (24) and projecting it onto the ansatz (37). The calculations become lengthy but pose no particular difficulty (see App. B).

## 4.3 Resolution of the flow equations

The ansatz (37) defines a set $\mathcal{L}$ of renormalisation functions

$$\mathcal{L} \equiv \left\{ f_{\kappa,n}^{\nu}, f_{\kappa,n}^{D}, f_{\kappa,n}^{\lambda}, \chi_{\kappa,n}^{a}, \chi_{\kappa,n}^{b}, \chi_{\kappa,n}^{c}, \gamma_{\kappa,n}^{a}, \gamma_{\kappa,n}^{b}, \gamma_{\kappa,n}^{c}, \mu_{\kappa,n}^{a}, \mu_{\kappa,n}^{b} \right\}. \tag{45}$$

In the following, we consider successive orders of approximation consisting in retaining, at a given order, only a subset $\mathcal{L}_\alpha$ composed of the $\alpha$ first functions of $\mathcal{L}$. We generically study $\mathcal{L}_3$, $\mathcal{L}_6$, $\mathcal{L}_9$ and $\mathcal{L}_{11} \equiv \mathcal{L}$ successively. The flow equations of the functions contained in the subset $\mathcal{L}_\alpha$ are integrated numerically, using standard procedures detailed in App. C, from the initial scale $\Lambda_{\mathrm{UV}}$ (respectively $\Lambda_{\mathrm{IR}}$ for the inverse RG flow) to the final scale $\Lambda_{\mathrm{IR}}$ (respectively $\Lambda_{\mathrm{UV}}$). Within all approximations, and both for the direct and the inverse flow, a fixed point is reached. We now present the results in details, for the direct flow in Sec. 5 and for the inverse flow in Sec. 6.

# 5 Results for the direct RG flow

In this section, the flow is integrated from the UV scale $\Lambda_{\mathrm{UV}}$ up to the IR scale $\Lambda_{\mathrm{IR}}$, starting from the initial condition (38) and (39), and zero for all the other functions. For $f_\kappa^D$, it turns out that we cannot directly set $D_{\mathrm{LR}} = 0$. This is because the SR part of the forcing is only non-zero at large scales (small wavenumbers), while the RG integration starts from large wavenumbers. For these wavenumbers, the initial condition for $f_\kappa^D$ is effectively zero in the absence of LR forcing, and it turns out that it is very difficult to stabilise the numerical integration starting from $f_D = 0$. Thus we keep a non-zero constant initial value for $f_\kappa^D$, which is equivalent to setting $\rho = 0$ in the LR part with a small $D_{\mathrm{LR}}$. We explain in the following that this constitutes an important drawback of the direct RG procedure, which prevents us from fully accessing the SR fixed point. This issue is cured by implementing an inverse RG in Sec. 6. On the other hand, the direct RG is well-suited to study turbulence generated by a LR power-law self-similar forcing, which we now investigate.

## 5.1 Fixed point with self-similar forcing

We start by presenting the results for $\rho = 0$. We obtain a fixed point within all approximation orders. We show in Fig. 1 the evolution with the RG scale of the renormalisation functions within the $\mathcal{L}_9$ order, *i.e.* including all the vertices up to cubic order in the fields. We observe that all the renormalisation functions gradually deform during the flow, starting from the large wavenumbers down to the small ones, until a fixed value is attained for all the shells, once the RG scale has crossed the integral shell. We define the scaling exponents of the quadratic functions in the inertial range as

$$f_{\kappa,n}^{\nu} \sim k_n^{\eta_\nu}, \quad f_{\kappa,n}^{D} \sim k_n^{\eta_D}, \quad f_{\kappa,n}^{\lambda} \sim k_n^{\eta_\lambda}. \tag{46}$$

We find, within all approximations, the values $\eta_\nu = 2/3$, $\eta_D = \eta_\lambda = 0$ up to numerical precision, which indicates that only $f_\kappa^\nu$ acquires a scaling different from its initial one $\eta_\nu = 2$. This in turn yields K41 scaling. Indeed, one can infer from the expression of $\Gamma_{2,\kappa}$ in (37), upon inserting the scaling (46), the scaling dimensions of the frequency and of the fields. One obtains

$$\omega \sim k_n^{\eta_\nu - \eta_\lambda} = k_n^{2/3}, \quad \bar{u}_n \sim k_n^{(\eta_\nu - \eta_\lambda - \eta_D)/2} = k_n^{1/3}, \quad u_n \sim k_n^{-(\eta_\nu - \eta_\lambda - \eta_D)/2} = k_n^{-1/3}, \tag{47}$$

which corresponds to K41 scaling. Pursuing the analysis of the terms $\Gamma_{3,\kappa}^{uu\bar{u}}$ and $\Gamma_{3,\kappa}^{u\bar{u}\bar{u}}$, and inserting the previous scaling, one obtains for the cubic vertices $\gamma_{\kappa,n}^{X} \sim k_n^{0}$ and $\chi_{\kappa,n}^{X} \sim k_n^{1/3}$ for

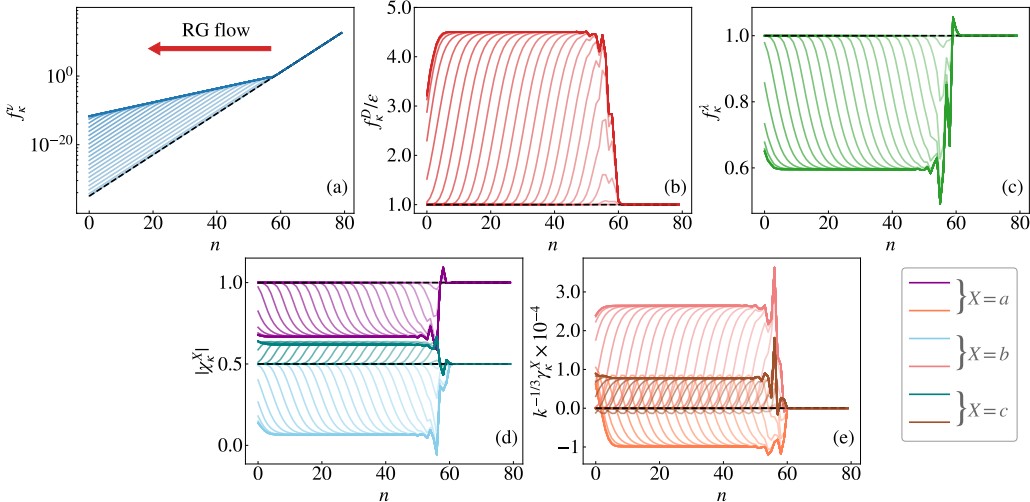

Figure 1: Evolution of the renormalisation functions at order $\mathcal{L}_9$ during the direct (from UV to IR) RG flow, starting from the initial condition at $\Lambda_{\text{UV}}$ represented by a dashed line. All the functions attain a fixed form at the end of the flow when $\kappa = \Lambda_{\text{IR}}$, indicated by a bold line.

$X = a, b, c$. This is precisely the behaviour observed in Fig. 1(d) and (e). We find that the quartic vertices also conform to K41 scaling, that is $\mu_{\kappa,n}^{X} \sim k_n^{2/3}$. Since the structure functions are reconstructed from the vertices and propagator as shown in (32)–(34), one expects that they also inherit K41 scaling [2], given by

$$S_p(k_n) \sim k_n^{p(\eta_D - 2\eta_\nu - \eta_\lambda + 1) + 3\eta_\nu + \eta_\lambda - \eta_D - 2} = k_n^{-p/3}. \tag{48}$$

This can be checked numerically. The fixed-point profiles of all the renormalisation functions are recorded, from which the renormalised propagator and vertices are deduced. They are then used to compute the structure functions following the definitions given in Sec. 3.4. The result is presented in Fig. 2(a). We obtain that the structure functions exhibit a wide inertial range of power-law behaviour, with exponents which coincide, up to numerical precision, with the K41 scaling, *i.e.* $\zeta_p = p/3$. Thus, this fixed point is not anomalous. We checked that this result is very robust and persists at all orders of approximations, and when varying the parameters. We argue in the following that this fixed point corresponds to the LR fixed point found in perturbative RG.

## 5.2 Long-Range fixed point and crossover to the Short-Range one

In order to make the link with perturbative results, we consider a purely LR forcing, *i.e.* we set $D_{\text{SR}} = 0$, and we vary the value of $\rho$ from $\rho = -2$ to $\rho = 1$. Again, we find a fixed point within all approximations for all values of $\rho$, from which we can compute the structure functions. The result for $S_2$ is displayed in Fig. 2(b), which shows that it behaves as a power law in the whole inertial range $S_2 \sim k_n^{-\zeta_2^{\text{LR}}}$ with an exponent

$$\zeta_2^{\text{LR}} = \frac{2}{3}(1 + \rho). \tag{49}$$

---

[2] The precise statement is that the scaling dimensions (in the RG scale $\kappa$) of the vertices are fixed by the Ansatz. The behaviour is wavenumber $k_n$ is then expected to be fixed by the scaling dimensions if the flow equations are decoupling at large momentum (see [27, 28] for a complete discussion of this point). The decoupling is automatically satisfied in shell models due to the locality in shells of the flow equations (at least for local vertices).

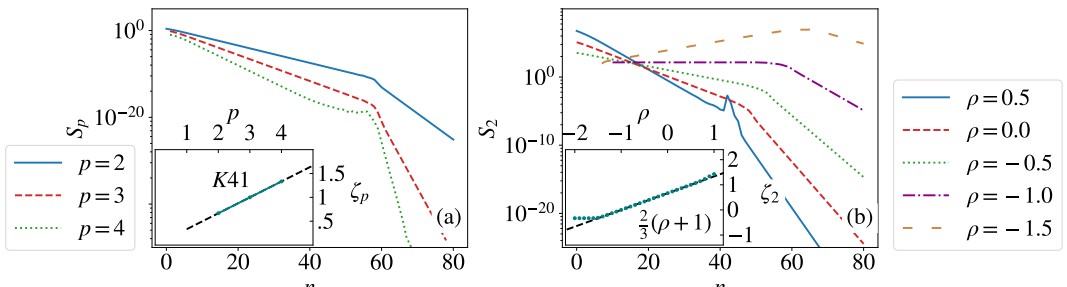

Figure 2: Structure functions obtained in the direct RG flow, starting from an initial condition $f^D_{\lambda,n} = D_{\text{LR}} k_n^{-\rho}$. (a) Structure functions $S_p$ with $p = 2, 3, 4$ for $\rho = 0$, with their exponents $\zeta_p$ as a function of $p$ in inset. This corresponds to K41 scaling. (b) Second order structure function $S_2$ for different values of $\rho$, with their exponents $\zeta_2$ as a function of $\rho$ in the inset. The observed scaling $\zeta_2^{\text{LR}} = 2(1 + \rho)/3$ is not anomalous.

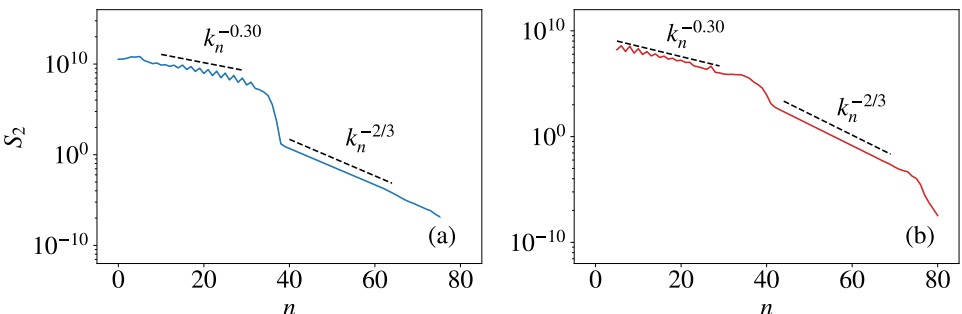

Figure 3: Interplay between the LR and SR forcing for high values of $r$, in the direct RG flow (a), and in the inverse RG flow (b), within the $\mathcal{L}_3$ approximation. The large wavenumbers $k_c \lesssim k_n \ll \eta^{-1}$ are controlled by the LR fixed point, whereas the small wavenumbers $k_0 \lesssim k_n \ll k_c$ are controlled by the SR fixed point, where $k_c$ roughly corresponds to the shell $n = 35$. The $\mathcal{L}_3$ approximation is chosen because it yields a value $\zeta_2 \simeq 0.3$ clearly distinct from the K41 one $\zeta_2 = 2/3$, although far from the expected value from numerical simulations.

The exponent (49) corresponds to a normal (dimensional) scaling, with no anomalous corrections. Hence the purely LR fixed point is non-intermittent. Let us notice that the behaviour of $S_2$ in the dissipation range can be derived from the large $k_n$ limit of the flow equations (see App. B). One finds $S_2 \sim k_n^{-(2+\rho)}$, which precisely describes the behaviour of the large shell indices in Fig. 2(b).

We find that the LR fixed point is not anomalous. This includes in particular the case $\rho = 0$. A question arises whether this fixed point is the same as the one which controls turbulence generated by a large-scale forcing. As explained at the beginning of the section, we cannot remove the LR component of the forcing because of numerical stability issues. However, we can study the effect of increasing the ratio $r = D_{\text{SR}}/D_{\text{LR}}$. We show in Fig. 3(a) the result for $S_2$ obtained for the larger value of $r$ we could attain. We observe the emergence of another scaling regime for scales $k_n \ll k_c$ where $k_c$ is the crossover scale from a regime where the LR component of the forcing dominates to a regime where the SR component dominates. The important point is that this new scaling regime features an exponent which is different from the K41 one $\zeta_2 = 2/3$. This evidences that the LR and the SR fixed points are different, and

induce different scaling. In order to confirm this finding, we now turn to the study of the purely SR fixed point. The direct RG flow does not permit this study, but it can be done within the inverse RG, which is presented in Sec. 6. Prior to this, let us elaborate on the consequences of this finding fro hydrodynamic turbulence.

### 5.3 Discussion on the analogy with 3D hydrodynamic turbulence

The early implementations of the RG relied on a perturbative expansion in the renormalised coupling, which is controlled by the distance to an upper critical dimension $\varepsilon = d_c - d$ [4]. However, for the Navier-Stokes equation, there is no upper critical dimension, and the interaction is the advection term, whose coupling is not adjustable (it is 1). As mentioned in the introduction, the strategy to artificially introduce a small parameter has been to consider a LR forcing with a power-law spectrum [15]:

$$N(k) \propto k^{4-d-\varepsilon}. \tag{50}$$

This self-similar forcing yields in perturbative RG a power spectrum $E(k) \propto k^{1-2\varepsilon}$, and a second order structure function scaling as $S_2(\ell) \propto \ell^{2\varepsilon/3-2}$ [7]. Hydrodynamic turbulence corresponds to $\varepsilon \to 4$, for which the convergence of the perturbative expansion is questionable. Moreover, a freezing should occur for values $\varepsilon > 4$ to recover universality, but such a freezing mechanism is absent from the pertubative treatment [7]. Since in the framework of perturbative RG, the physical situation can only be accessed as this precarious limit from the LR forcing, the question arises as to which extent it correctly describes hydrodynamic turbulence. In other words, are the multi-fractal behaviour of turbulence and the intermittency exponents universal with respect to *any* forcing, including a LR one? Since this is beyond the reach of perturbative RG, this question was investigated using direct numerical simulations in 3D [16–18]. They clearly evidenced that for $\varepsilon < 4$, the probability distribution of the velocity increments is Gaussian, and the exponents of the structure functions conform with the perturbative prediction, while for $\varepsilon > 4$, the distribution develops the large tails and an intermittent behaviour expected for a large-scale forcing, with anomalous exponents for the structure functions. However, let us notice that in the simulations, the power-law forcing is regularised, which is equivalent to the co-existence of both a LR and a SR component. Let us mention that in contrast, in 2D turbulence, numerical simulations suggest that the dimensional scaling predicted by perturbative RG is never observed, as it is always sub-dominant compared with the dominant scaling in either the direct or the inverse cascade [19, 20]. In this work, we only focused on parameters for which the shell model mimics 3D turbulence.

For shell models, the limit $\rho \to 0$ corresponds to the limit $\varepsilon \to 4$ in 3D hydrodynamic turbulence. Indeed, since the Sabra shell model is zero-dimensional, using (50), we can infer the relation between $\rho$ and $\varepsilon$, yielding $\rho = \varepsilon - 4$. Our result for $S_2$ at the LR fixed point hence coincides with the scaling obtained at the pertubative fixed point, *i.e.* $S_2 \propto k_n^{-2/3(\rho+1)} = k_n^{2-2\varepsilon/3}$. Our analysis shows that for the shell models, the LR forcing, yielding similar result as the perturbative analysis in 3D Navier-Stokes, and the SR forcing, correspond to two distinct universality classes. While the SR fixed point is anomalous, the LR one is not, it is characterised by K41 dimensional scaling. Our results are thus compatible with results from 3D numerical simulations. Moreover, for a purely LR forcing, the energy spectrum does not freeze for $\rho < 0$. Such a freezing can only occur in the presence of the SR forcing, which takes over and whose spectrum is independent of $\rho$. As a consequence, the anomalous exponents of the SR fixed point cannot be inferred from the $\rho \to 0$ limit of the LR fixed point. It would be interesting to study within the FRG if the same phenomenon occurs for the Navier-Stokes equations.

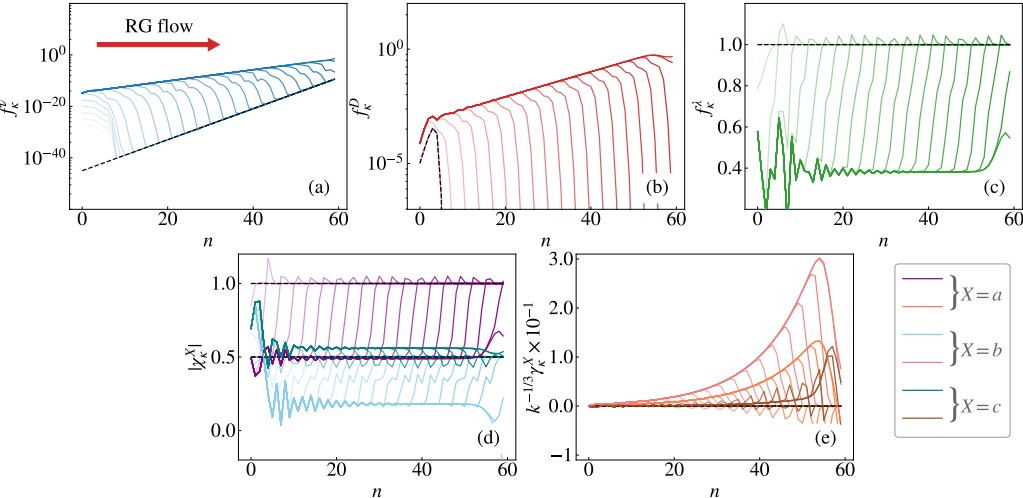

Figure 4: Evolution of the renormalisation functions at order $\mathcal{L}_9$ during the inverse (from IR to UV) RG flow, starting from the initial condition at $\Lambda_{\text{IR}}$ represented by a dashed line. All the functions attain a fixed form at the end of the flow when $\kappa = \Lambda_{\text{UV}}$, indicated by a bold line.

## 6 Results for the inverse RG flow

In this section, the flow is integrated from the IR scale $\Lambda_{\text{IR}}$ down to the UV scale $\Lambda_{\text{UV}}$, starting from the initial condition (38) and (39), and zero for all the other renormalisation functions. Within the inverse RG, we can safely set $D_{\text{LR}} = 0$ since the flow starts at the large scales (small wavenumbers) where $N_{L,n}$ is finite. Hence the numerical integration is not impeded by instabilities. We thus determine the statistical properties at the SR fixed point and show that it is anomalous.

### 6.1 Fixed point with large-scale forcing

As for the direct RG flow, we obtain a fixed point within all approximation orders. We show in Fig. 4 the evolution with the RG scale of the fixed-point functions within the same $\mathcal{L}_9$ order as for Fig. 1. In the inverse RG, both the functions $f_\kappa^\gamma$ and $f_\kappa^D$ acquire a non-trivial scaling at the fixed point. Comparing with Fig. 1, one can observe that this scaling builds up from the large scales towards the smaller scales. It nicely follows the cascade picture expected in physical space. We compute from the fixed-point functions the structure functions, which are displayed in Fig. 5. They exhibit a large inertial range with power-law behaviour, but with exponents $\zeta_p$ that do not correspond to the K41 result. The inset of Fig. 5 shows the spectrum of $\zeta_p$ obtained within the $\mathcal{L}_9$ order. We find that all the exponents are anomalous $\zeta_p \neq p/3$. However, the spectrum is linear, rather than convexe as expected from multi-fractality. Hence, this scaling is similar to that of a $\beta$-model [51], *i.e.* this reflects an anomalous but unifractal scaling.

We now comment on the robustness of our results. The anomalous and unifractal behaviour is obtained within all orders of approximation studied in this work. However, the precise value of the correction varies with the order. We present in Table 1 the values obtained at the successive orders $\mathcal{L}_3$, $\mathcal{L}_6$, $\mathcal{L}_9$ and $\mathcal{L}$. At the maximal bi-quadratic level $\mathcal{L}$, the exponent $\zeta_2 \simeq 0.75$ is not too far from the value $\zeta_2 \simeq 0.720 \pm 0.008$ obtained from numerical simulations of the Sabra model [45]. Moreover, the exponent $\zeta_3$ is compatible with the exact result

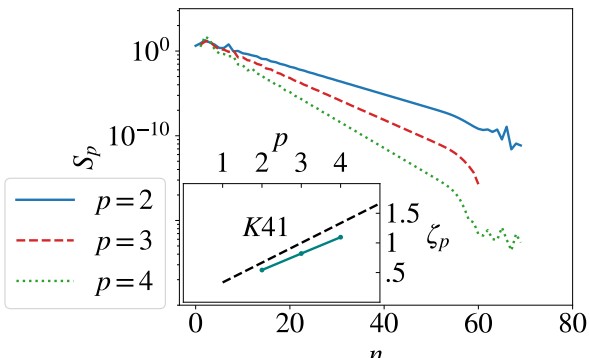

Figure 5: Structure functions $S_p$ with $p = 2, 3, 4$ obtained in the inverse RG flow, starting from an initial condition (38) with $D_{\text{LR}} = 0$. The exponents $\zeta_p$ as a function of $p$ are shown in inset. This corresponds to anomalous, but unifractal, scaling.

|         | $\zeta_2$         | $\zeta_3$       | $\zeta_4$          |
|---------|-------------------|-----------------|--------------------|
| $\mathcal{L}_3$ | 0.29              | 0.44            | 0.58               |
| $\mathcal{L}_6$ | 0.54              | 0.82            | 1.09               |
| $\mathcal{L}_9$ | 0.55              | 0.82            | 1.09               |
| $\mathcal{L}$   | 0.75              | 1.13            | 1.51               |
| num.    | $0.72 \pm 0.008$  | $1.0 \pm 0.005$ | $1.256 \pm 0.012$  |

Table 1: Exponents $\zeta_p$ of the $p^{\text{th}}$ order structure functions $S_p(k_n)$ obtained in the inverse RG flow at successive orders of approximations within the vertex expansion presented in Sec. 4.

$\zeta_3 = 1$. However, the convergence is slow, and there is still a relative difference of about 25% on the exponents between the orders $\mathcal{L}_9$ and $\mathcal{L}$. While this result on its own is clearly not sufficient to discard K41 scaling, we can show that this fixed point is distinct from the K41 one.

For this, we study as in the direct RG flow the interplay between the SR and LR fixed point. In the inverse flow, it is the limit $r \to 0$ (purely LR forcing) which is difficult to attain for numerical stability issues, but it is still possible to approach it. We show in Fig. 3(b) the result for $S_2$ for the smallest value of $r$ we could reach, within the $\mathcal{L}_3$ approximation. It clearly confirms the existence of the two different fixed points, which yield two different scaling regimes. Indeed, the dimensional value $\zeta_2 = 2/3$ is found within all approximations at the LR fixed point, both in the direct and in the inverse RG flow. In contrast, the value of $\zeta_2$ at the SR fixed point varies with the order of the approximation and, although it converges slowly, it is distinct from 2/3.

## 6.2  Discussion on possible improvements

The convergence within the vertex expansion implemented here is not fully satisfactory. One could in principle include more vertices in order to refine the estimate for the $\zeta_p$. However, we believe that this would not be sufficient to account for multi-fractality. Indeed, if the additional vertices remain local, in the sense that the associated renormalisation functions depend only on one wavenumber $k_n$, one is bound to obtain a unifractal spectrum of anomalous exponents $\zeta_p$. In fact, once the scaling of the fields and of the frequency is fixed, through the quadratic

terms, it determines the scaling of all other possible local vertices.

To circumvent this, one could introduce non-local vertices, *i.e.* vertices with associated renormalisation functions depending on more than one wavenumbers. This is possible starting from the 4-point vertices. As shown in App. A, configurations such as $\bar{u}^*_{n_1} \bar{u}^*_{n_2} u_{n_1} u_{n_2}$ are compatible with the conservation of quasi-momentum, and they could be associated with two-wavenumber renormalisation functions $\mu_\kappa(k_{n_1}, k_{n_2})$. Alternatively, one could let the renormalisation functions depend on frequency, as well as wavenumbers. This would probably require to also regulate the frequency sector.

Finally, another promising route would be to compute the structure functions from composite operators, rather than reconstructing them from the vertex functions. Indeed, in shell models, the structure functions involve coinciding times, and one could consider the associated composite operators made of products of velocities at a single time. All these studies are left for future work.

## 7   Conclusion and perspectives

We presented the FRG formalism to study shell models of turbulence. In particular, we showed how to implement both a direct (conventional) RG, and an inverse one, which had been for long advocated in the context of turbulence but hardly ever realised. We applied this framework to study the Sabra shell models, within an approximation scheme called the vertex expansion, up to bi-quadratic order in the fields. This allowed us to study the effect of the range of the forcing, from a SR forcing concentrated at large scales, to a LR self-similar one $\propto k_n^{-\rho}$ exerted at all scales. We showed that the LR and SR forcing yield two distinct fixed points. The LR one is characterised by K41 scaling, with the energy spectrum continuously depending on the exponent $\rho$, while the SR one exhibits a different scaling, which is anomalous. Within the approximation scheme of this work, we obtained only a uni-fractal behaviour for the exponents $\zeta_p$ of the structure functions, rather than the multi-fractal one. We discussed possible improvements to overcome this limitation. Of course, the theoretical understanding of the fundamental ingredients which lead to multi-fractality in shell models is of paramount importance, since these findings could shed some light on analogous mechansims in hydrodynamic turbulence.

Let us make a final remark concerning the numerical cost. The integration of the flow equations presented in this work is fast. It is in particular much faster than direct numerical simulations of the shell model since one does not have to perform any averages: the deterministic solution of the FRG flow equation directly provides the averaged statistical properties of the system. The improvement of the approximation, through for instance the inclusion of non-local vertices, would then lead to longer computational time to solve the associated FRG equations. It would be extremely interesting to test which level of accuracy on the anomalous exponents can be reached within a still competitive approximation scheme compared to direct numerical simulation.

## Acknowledgement

The authors warmly thank N. Wschebor and B. Delamotte for stimulating discussions, and are very grateful to G. Eyink for very relevant comments on our work and for the suggestion of the inverse RG. L.C. acknowledges support from the French ANR through the project NeqFluids (grant ANR-18-CE92-0019) and support from Institut Universitaire de France (IUF). M.T. and F.B. acknowledge support from the Simons Foundation, grant number 663054, through a

subagreement from the Johns Hopkins University. The contents of this publication are solely the responsibility of the authors and do not necessarily represent the official views of Simons Foundation or the Johns Hopkins University.

# A Constraints from translational invariance on 4-point configurations

In this appendix, we provide a complete classification of the 4-point configurations permitted by the conservation of the quasi-momentum. To begin, let us recall the definitions: to each shell variable (respectively their complex conjugate) of shell index $n$ we associate the quasi-momentum $\kappa_n = \varphi^n$ (resp. $\kappa_n = -\varphi^n$), where

$$\varphi = \frac{1 + \sqrt{5}}{2},\tag{51}$$

is the golden mean. A generic $p$-point configuration is thus given by the data of $p$ shell indices $n_1, \ldots n_p \in \mathbb{Z}$ and $p$ conjugation indices $\epsilon_1, \ldots \epsilon_p \in \{-1, 1\}$. A configuration is said to conserve quasi-momentum if the sum of its signed quasi-momenta is zero, or in other words

$$\sum_{i=1}^{p} \epsilon_i \varphi^{n_i} = 0.\tag{52}$$

The following of the appendix is divided in two parts. First, we derive a general formula for a quasi-momentum conserving configuration in terms of an integer polynomial. Then, we apply it to classify all 4-point quasi-momentum conserving configurations.

## A.1 General formula

In preparation of the main statement, we have to reduce some redundancies present in the definition above. First, we observe that the quasi-momentum conservation condition is symmetric in the shell variables: any permutation $\left(\epsilon_{\sigma(i)}, n_{\sigma(i)}\right)_{1 \leq i \leq p}$ of a quasi-momentum conserving configuration $(\epsilon_i, n_i)_{1 \leq i \leq p}$ is also quasi-momentum conserving. This means that we can restrict our attention to equivalence classes of configurations up to permutations. Second, when two shell variables in a configuration have the same shell index and opposite conjugation indices, their quasi-momenta cancel each other. We call such case a trivial cancellation. Given a $p$-point quasi-momentum conserving configuration, one can construct a $(p+2)$-point quasi-momentum conserving configuration by adding a trivial cancellation. In the following, we thus consider only configurations which do not contain trivial cancellations.

There is a one-to-one mapping between equivalence class of configurations without trivial cancellations and the integer polynomials $\mathbb{Z}[X]$. Explicitly, each coefficient of $P = a_D X^D + \cdots + a_0$ has its sign (resp. its absolute value) given by the conjugation index (resp. the multiplicity) of the shell variables at the shell index $n$. In particular, let us notice that the $\ell_1$-norm of the polynomial is equal to the number of points in the configuration (recalling that we discarded trivial cancellations):

$$\|P\|_1 = \sum_{i=0}^{D} |a_i| = p\,.\tag{53}$$

By the above mapping, the quasi-momentum conserving configurations (up to permutations, without trivial cancellations) are in one-to-one correspondence with the ideal in $\mathbb{Z}[X]$ of the polynomials having $\varphi$ as root. It turns out that the latter is actually a principal ideal in $\mathbb{Z}[X]$, generated by the minimal polynomial of $\varphi$, namely $X^2 - X - 1$. To show this, first

observe that it is true when viewed as an ideal in the principal ideal domain $\mathbb{Q}[X]$ [58]. Thus, for any $P = a_D X^D + \cdots + a_0 \in \mathbb{Z}[X]$ satisfying $P[\varphi] = 0$, there exists a $Q = b_{D-2}X^{D-2} + \ldots + b_0 \in \mathbb{Q}[X]$ such that

$$P = (X^2 - X - 1)Q. \tag{54}$$

Then, solving the recurrence relations

$$\forall 0 \leq n \leq D, \qquad a_n = b_{n-2} - b_{n-1} - b_n, \tag{55}$$

(with the understanding that in the above expression $b_{n>D-2} = b_{n<0} = 0$), we find that

$$\forall 0 \leq n \leq D-2, \qquad b_n = \sum_{k=0}^{n}(-1)^{n-k+1}F_{n-k+1}a_k, \tag{56}$$

where the $F_n$'s are the Fibonacci numbers ($F_1 = F_2 = 1$ and $F_n = F_{n-1} + F_{n-2}$). In particular, we deduce that in fact $Q \in \mathbb{Z}[X]$. In conclusion, the formula (54) with $Q \in \mathbb{Z}[X]$ is the most general formula such that the associated configuration is quasi-momentum conserving. Using the solution (56), we find that (54) is equivalent to the following conditions on the coefficients of $P$:

$$\begin{cases} \sum_{n=0}^{D-1}(-1)^{D-n}F_{D-n}a_n = 0, \\ \sum_{n=0}^{D}(-1)^{D+1-n}F_{D+1-n}a_n = 0, \end{cases} \tag{57}$$

## A.2   Application to 4-point configurations

We want to find all $P \in \mathbb{Z}[X]$ with $\|P\|_1 = 4$ such that (54) holds. We can understand the constraint on the $\ell_1$-norm by asking that the $|a_D|, \ldots, |a_0|$ realize a partition of 4. From (57), we deduce readily that partitions into one or two integers are prohibited. We are thus left with either $4 = 2 + 1 + 1$ or $4 = 1 + 1 + 1 + 1$. Furthermore, the only polynomials of degree $D = 2$ satisfying (54) are the polynomials $m(X^2 - X - 1)$, $m \in \mathbb{Z}$ of norm $3m$ so that for the 4-point configurations, we have $D \geq 3$. Without loss of generality, let us assume that $a_0 \neq 0$ and that $a_D > 0$. This amounts in the configuration to fixing the smallest shell index and the conjugation index of the shell variable with largest shell index.

Let us consider first the case $4 = 2 + 1 + 1$. Then, evaluating $P = a_D x^D + a_n X^n + a_0$ at $\varphi$ and its conjugate root $-\varphi^{-1}$, we get

$$\begin{cases} a_D \varphi^D + a_n \varphi^n + a_0 = 0, \\ a_D(-\varphi)^{-D} + a_n(-\varphi)^{-n} + a_0 = 0, \end{cases} \tag{58}$$

Using the triangle inequality, we have in particular that

$$\begin{cases} |a_D| \leq |a_n|\varphi^{n-D} + |a_0|\varphi^{-D}, \\ |a_0| \leq |a_n|\varphi^{-n} + |a_D|\varphi^{-D}. \end{cases} \tag{59}$$

As $\varphi^{-n}, \varphi^{n-D}, \varphi^{-D} < 1$, the choice $|a_D| = 2$ (respectively $|a_0| = 2$) cannot be a solution of the first line (resp. the second line), such that we are left with $|a_D| = 1, |a_0| = 1, |a_n| = 2$. Inserting back theses values and isolating the first term of the *r.h.s.* in the above inequalities, we obtain

$$\begin{cases} D - n \leq -\log_\varphi \frac{1-\varphi^{-D}}{2}, \\ n \leq -\log_\varphi \frac{1-\varphi^{-D}}{2} \end{cases} \tag{60}$$

where $\log_\varphi$ is the logarithm base $\varphi$. The *r.h.s.* is strictly decreasing and equal to 2 at $D = 3$. Consequently, the choice $D \geq 4$ leads to a contradiction and the only remaining candidate solutions have $D = 3$ and either $n = 1$ or $n = 2$. We verify that both cases are actual solutions of (57), with polynomials given respectively by

$$\begin{cases} n = 1 : & P = X^3 - 2X - 1 \\ n = 2 : & P = X^3 - 2X^2 + 1 \,. \end{cases} \tag{61}$$

Second, let us turn to the case $4 = 1 + 1 + 1 + 1$. As a preliminary, we remark that the only polynomials of degree $D = 3$ which are multiples of $X^2 - X - 1$ and have $|a_D| = |a_0| = 1$ are the ones found above, such that we can assume $D \geq 4$ in the following. Evaluating $P = a_D X^D + a_m X^m + a_n X^n + a_0$ at $\varphi$ and $-\varphi^{-1}$, we get

$$\begin{cases} a_D \varphi^D + a_m \varphi^m + a_n \varphi^n + a_0 = 0 \,, \\ a_D (-\varphi)^{-D} + a_m (-\varphi)^{-m} + a_n (-\varphi)^{-n} + a_0 = 0 \,. \end{cases} \tag{62}$$

Using triangle inequalities, the above equations entail

$$\begin{cases} 1 - \varphi^{-D} \leq \varphi^{m-D} + \varphi^{n-D} \leq 2\varphi^{m-D} \,, \\ 1 - \varphi^{-D} \leq \varphi^{-m} + \varphi^{-n} \leq 2\varphi^{-n} \,, \end{cases} \tag{63}$$

from which we deduce respectively that $m = D - 1$ and $n = 1$. Inserting back these values into the inequalities, the only candidate solution is $D = 4$ and indeed we find the following solution of (57):

$$P = X^4 - X^3 - X - 1 \,. \tag{64}$$

To sum up, we illustrate our results by listing all 4-point configurations of the shell variable $v$ (up to a global conjugation and up to shifting all shell indices by the same value) which preserve the quasi-momentum. The first and second line correspond respectively to polynomials of degree 3 and 4 obtained above, and the third line corresponds to the trivial cancellations.

$$\begin{gathered} \left\langle v_3 (v_1^*)^2 v_0^* \right\rangle, \quad \left\langle v_3 (v_2^*)^2 v_0 \right\rangle, \\ \left\langle v_4 v_3^* v_1^* v_0^* \right\rangle, \\ \forall n \in \mathbb{Z}, \left\langle v_n v_n^* v_0 v_0^* \right\rangle, \end{gathered} \tag{65}$$

## B   FRG flow equations for the quadratic functions

The calculation of the flow equations for the quadratic renormalisation functions $f_{\kappa,n}^v$, $f_{\kappa,n}^D$, and $f_\kappa^\lambda$ is straightforward. They can be deduced using the definition (44) from the flow equations for the two-point functions $\Gamma_\kappa^{(2)}$ which are given by (24). The trace in this equation can be computed using the explicit expressions (41), (41) and (43) for the 3- and 4-point vertices and (29) for the propagator $G_\kappa$. Since within the approximation (37), the renormalisation functions do not depend on the frequency, the only frequency dependence is the bare one present in the propagator, so that the frequency integral can be analytically carried out. For

instance, at order $\mathcal{L}_6$, one obtains:

$$
\partial_s f_{\kappa,n}^\nu = \frac{k_0^2 \lambda^{2n}}{2} \bigg[ \lambda^2 \chi_{\kappa,n+1}^a \chi_{\kappa,n+1}^b \Phi_\kappa^\nu(n+2,n+1) + \chi_{\kappa,n}^a \chi_{\kappa,n}^b \Phi_\kappa^\nu(n+1,n-1)
$$
$$
+ \lambda^2 \chi_{\kappa,n+1}^a \chi_{\kappa,n+1}^c \Phi_\kappa^\nu(n+1,n+2) + \frac{1}{\lambda^2} \chi_{\kappa,n-1}^a \chi_{\kappa,n-1}^c \Phi_\kappa^\nu(n-1,n-2)
$$
$$
+ \chi_{\kappa,n}^b \chi_{\kappa,n}^c \Phi_\kappa^\nu(n-1,n+1) + \frac{1}{\lambda^2} \chi_{\kappa,n-1}^b \chi_{\kappa,n-1}^c \Phi_\kappa^\nu(n-2,n-1) \bigg] \tag{66}
$$

$$
\partial_s f_{\kappa,n}^\lambda = \frac{k_0^2 \lambda^{2n}}{2} \bigg[ \lambda^2 \chi_{\kappa,n+1}^a \chi_{\kappa,n+1}^b \Phi_\kappa^\lambda(n+2,n+1) + \chi_{\kappa,n}^a \chi_{\kappa,n}^b \Phi_\kappa^\lambda(n+1,n-1)
$$
$$
+ \lambda^2 \chi_{\kappa,n+1}^a \chi_{\kappa,n+1}^c \Phi_\kappa^\lambda(n+1,n+2) + \frac{1}{\lambda^2} \chi_{\kappa,n-1}^a \chi_{\kappa,n-1}^c \Phi_\kappa^\lambda(n-1,n-2)
$$
$$
+ \chi_{\kappa,n}^b \chi_{\kappa,n}^c \Phi_\kappa^\lambda(n-1,n+1) + \frac{1}{\lambda^2} \chi_{\kappa,n-1}^b \chi_{\kappa,n-1}^c \Phi_\kappa^\lambda(n-2,n-1) \bigg] \tag{67}
$$

$$
\partial_s f_{\kappa,n}^D = \frac{k_0^2 \lambda^{2n}}{2} \bigg[ (\chi_{\kappa,n+1}^a)^2 \lambda^2 \Phi_\kappa^D(n+1,n+2)
$$
$$
+ (\chi_{\kappa,n}^b)^2 \Phi_\kappa^D(n-1,n+1) + \frac{(\chi_{\kappa,n-1}^c)^2}{\lambda^2} \Phi_\kappa^D(n-2,n-1) \bigg] \tag{68}
$$

where we have defined

$$
\Phi_\kappa^\nu(i,j) = \frac{f_{\kappa,i}^D}{\tilde{f}_{\kappa,i}^\nu (\tilde{f}_{\kappa,i}^\nu f_{\kappa,j}^\lambda + \tilde{f}_{\kappa,j}^\nu f_{\kappa,i}^\lambda)^2} \left( \dot{R}_{\kappa,j} f_{\kappa,i}^\lambda + \dot{R}_{\kappa,i} f_{\kappa,j}^\lambda \left( 2 + \frac{f_{\kappa,i}^\lambda \tilde{f}_{\kappa,j}^\nu}{f_{\kappa,j}^\lambda \tilde{f}_{\kappa,i}^\nu} \right) \right)
$$

$$
\Phi_\kappa^\lambda(i,j) = \frac{f_{\kappa,i}^\lambda f_{\kappa,j}^\lambda}{\tilde{f}_{\kappa,i}^\nu (\tilde{f}_{\kappa,i}^\nu f_{\kappa,j}^\lambda + \tilde{f}_{\kappa,j}^\nu f_{\kappa,i}^\lambda)^2} \frac{2\dot{R}_{\kappa,j} f_{\kappa,i}^D f_{\kappa,i}^\lambda + \dot{R}_{\kappa,i} f_{\kappa,i}^D f_{\kappa,j}^\lambda \left( 3 + \frac{f_{\kappa,i}^\lambda \tilde{f}_{\kappa,j}^\nu}{f_{\kappa,j}^\lambda \tilde{f}_{\kappa,i}^\nu} \right)}{\tilde{f}_{\kappa,i}^\nu f_{\kappa,j}^\lambda + \tilde{f}_{\kappa,j}^\nu f_{\kappa,i}^\lambda}
$$

$$
\Phi_\kappa^D(i,j) = -\frac{f_{\kappa,i}^D f_{\kappa,j}^D \left( \dot{R}_{\kappa,j} f_{\kappa,i}^\lambda \left( 2 + \frac{\tilde{f}_{\kappa,i}^\nu f_{\kappa,j}^\lambda}{\tilde{f}_{\kappa,j}^\nu f_{\kappa,i}^\lambda} \right) + \dot{R}_{\kappa,i} f_{\kappa,j}^\lambda \left( 2 + \frac{\tilde{f}_{\kappa,j}^\nu f_{\kappa,i}^\lambda}{\tilde{f}_{\kappa,i}^\nu f_{\kappa,j}^\lambda} \right) \right)}{\tilde{f}_{\kappa,i}^\nu \tilde{f}_{\kappa,j}^\nu (f_{\kappa,j}^\lambda \tilde{f}_{\kappa,i}^\nu + f_{\kappa,i}^\lambda \tilde{f}_{\kappa,j}^\nu)^2} \tag{69}
$$

and introduced the following notations for the function:

$$
\tilde{f}_{\kappa,n}^\nu = f_{\kappa,n}^\nu + R_{\kappa,n} , \tag{70}
$$

and for the derivative of the regulator:

$$
\dot{R}_{\kappa,n} = -2\nu k_n^2 \left( \frac{k_n}{\kappa} \right)^2 \partial_{x^2} r \left( \left( \frac{k_n}{\kappa} \right)^2 \right) . \tag{71}
$$

The RG time $s$ is defined as $s = \log \frac{\kappa}{\kappa_{\text{init}}}$ where $\kappa_{\text{init}} = \Lambda_{\text{UV}}$ or $\Lambda_{\text{IR}}$. Note that all these flow equations are real for real functions and regulators. Thus, if one starts at $\kappa = \kappa_{\text{init}}$ with real functions, they remain real along the flow. In higher-order approximations, these expressions become lengthy, and are not written explicitly here. However they are available in the form of a Mathematica file at this repository.

Let us notice that at the beginning of the direct RG flow for $\kappa \lesssim \Lambda_{\text{UV}}$, the renormalisation is very weak. In fact, for a pure LR forcing, the flow equations (66), (67), and (68) behave at

large $\kappa$, *i.e.* when the three functions are close to their initial condition, and $k_n \sim \kappa$ where the flows are maximal, as:

$$\partial_s f_{\kappa,n}^{\nu} \sim \frac{\nu \kappa^2}{(\eta \kappa)^{\rho+4}}, \qquad \partial_s f_{\kappa,n}^{D} \sim \frac{D^{LR} \kappa^{-\rho}}{(\eta \kappa)^{\rho+4}}, \qquad \partial_s f_{\kappa,n}^{\lambda} \sim \frac{1}{(\eta \kappa)^{\rho+4}}. \tag{72}$$

The right hand sides of these flow equations is suppressed at large $k_n$ by large powers of $\kappa \eta$. The flow becomes non-negligible when the RG scale reaches the order of the Kolmogorov scale $\eta^{-1}$. Since the flow is frozen by the regulator for all shells with $k_n \gg \kappa$, the shells in the dissipation range beyond this scale are almost not affected by the flow, and the three functions $f_\kappa^\nu$, $f_\kappa^D$ and $f_\kappa^\lambda$ for $k_n \gtrsim \eta^{-1}$ merely reflect their initial condition. As a consequence, $S_2$ behaves in the dissipation range as

$$S_2(k_n) = \frac{f_{0,n}^D}{2 f_{0,n}^\nu f_{0,n}^\lambda} \overset{n \gtrsim -\frac{\ln(k_0 \eta)}{\ln \lambda}}{\simeq} \frac{D^{\mathrm{LR}} k_n^{-\rho}}{2 \nu k_n^2} \sim k_n^{-(2+\rho)}. \tag{73}$$

This behaviour is clearly observed in Fig. 2(b) for large shell indices.

## C   Numerical integration of the flow equations

In this appendix, we provide the details of the numerical integration of the flow equations. At a given order of the vertex expansion, one considers a subset $\mathcal{L}_\alpha$ of the ensemble $\mathcal{L}$ of renormalisation functions entering the ansatz (37) with $\alpha$ denoting the number of functions in the subset. Their flow equations form a set of coupled ordinary differential equations. The chosen ODE integrator is a simple Euler scheme, with an adaptive time stepping. The time step is chosen so that the evolution of the flowing function in a single time step is smaller than one percent. The source code of our solver is available in this repository. Let us now specify the boundary conditions. Since the flow equations couple neighbouring shells and since we truncate at a finite number of shells, we need to prescribe the values of the functions on the shells at the external boundary of the integration grid. In order to do so, we assume a power law behaviour for the running functions at large wavenumbers. This is motivated by the expected behaviour at the fixed-point. Hence, we extrapolate each running function $f_\kappa^X$ for each RG step $\kappa$, using logarithmic finite differences at the boundary point

$$\zeta^X = \log f_{\kappa,N}^X - \log f_{\kappa,N-1}^X \tag{74}$$

and we fix the value of the sequence at $n > N$ following

$$f_{\kappa,n}^X = f_{\kappa,N}^X \lambda^{\zeta^X(n-N)}. \tag{75}$$

This principle is also used to extrapolate the value of the functions at shell indices smaller than $n = 0$. Since the flow is computed with fully dimensional quantities, we must also specify a starting and a finishing RG scale. For every run, we chose $\Lambda_{\mathrm{UV}} = 2^4 \times k_N$ and $\Lambda_{\mathrm{IR}} = 2^{-4} \times k_0$. We checked that this choice has no influence on the results presented in this paper. The precise value of the parameters used for the inverse and direct RG flows differ slightly, mostly for numerical stability reasons, but both correspond to $\lambda = 2$ and $c = -1/2$. The values of the parameters, along with the data used to produce the plots contained in this paper, may be found in the same repository as the source code of the numerical solver.

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
