# Peer review of "Functional renormalisation group approach to shell models of turbulence"

_SciPost Physics_

## Round 1 · Referee Report · Anonymous (Referee 1) · 2022-10-19

Strengths

1- clearly written 2-scientifically valid, state of the art application of functional renormalization methods to scaling analysis.

Weaknesses

-functional renormalization scaling predictions rely on closures. This is however not a weakness of the paper in itself rather of the method so far.

Report

The manuscript describes the application of functional RG to the ``Sabra'' shell model of turbulence. Shell models are stylized models of fluid turbulence. Before the proof of anomalous scaling in the Kraichnan methods, shell models possibly provided convincing evidences
though only numerical that anomalous scaling in passive scalar turbulence is not a finite effect and that the same may well be true for the non-linear advecting field.
The present paper introduces a closure, the so called LO approximations, of the functional renormalization flow which is then numerically solved.
The results are consistent within numerical accuracy with the prediction of anomalous scaling of the shell model counter-parts of structure functions
by direct numerical simulations. The authors took also care to verify that whereas the anomalous exponents are universal with respect to the bare forcing amplitude and molecular dissipation they may not be Fig 2 right panel with respect to the shell ratio and the shell model parameter.
Another interesting observation regards the dependence of the RG fixed point upon the type of forcing. The authors' finding is that there exist two distinct fixed points depending on the nature of the forcing. If the forcing is concentrated on one infra-red shell the fixed point describes anomalous scaling. If instead the forcing is a power-law on all shells the corresponding fixed point is associated to dimensional scaling. The authors speculate that the same phenomenon may occur in realistic models of fluid turbulence. This conclusion maybe, however, too hasty or perhaps not clearly formulated. Numerical simulations of the stochastic Navier-Stokes equation in two dimensions by A Mazzino et al PRL 99 (14), 144502, and JSTAT 2009 (10), P10012 give, so far undisputed evidence, that power law forcing may bring about both type of behaviors. Specifically, dimensional scaling dominates in parametric regions where it is expected to be so based on the analysis of the Karman-Howarth-Monin equation. Conversely inverse energy cascade for ($ d=2 $ and $ \varepsilon<4 $) and or direct enstrophy cascade ($ \varepsilon>6 $) become the dominant contributions to scaling in complementary parametric regions again in agreement with predictions based on the asymptotic analysis of the KHM equation. In my understanding, the benchmark for identifying a ``turbulence'' fixed point for functional RG in Navier-Stokes is the 5/3 law and consistency with Galilean invariance (Ward identities). As both types of forcing (e.g. power law at $ \epsilon=4 $ and large scale in $d=3$) satisfy these requirements, the interpretation of the fixed points they bring about in what should be the fully developed turbulence regime remains, in my opinion, inconclusive. In the case of shell models the absence of spatial structure maybe ultimately responsible for the observed phenomenon. A more careful discussion by the authors of this point would be desirable.

Overall the paper is well written and comes from a collaboration including one of the leading experts (Canet) on FRG methods so I have no remarks on the technical implementation. Hence my recommendation is to publish with minor amendments in relation to my comments above.

---

## Round 1 · Referee Report · Anonymous (Referee 2) · 2023-5-20

Strengths

Introduction of a powerful theoretical framework to describe hydrodynamic turbulence.
Captures anomalous scaling in shell model.
Able to contrast long-range and short-range forcing with an important consequence for existing attempts at describing turbulence within the perturbative renormalization group.
Manuscript is clearly presented.

Weaknesses

Lowest order approximation does not describe multi-fractal scaling

Report

In this manuscript the authors apply the functional Renormalization Group to simplified models of hydrodynamic turbulence known as shell models. They more specifically focus on one representative of the class of models, the Sabra shell model. The authors show for the first time the existence of a fixed point controlling the universal properties in the case of a forcing at large scale. They do so by approximating the exact functional RG flow equations through a truncation at the lowest order of the vertex expansion. The full functional dependence on wave-number is retained, which allows the authors to calculate the scaling dimensions of the structure functions. The anomalous character of the scaling that originates from intermittency is well captured and the exponent for the second-order structure function is is good agreement with computer simulation results. However, the present approximation fails at describing the expected multi-fractal scaling of the model.

An interesting property of the functional RG method is that it allows a study of the nature of forcing on the fixed point. Long-range forcing is defined as scale free while short-range forcing is concentrated at the large scale only. The authors show that the two types of forcing lead to distinct fixed points and, more importantly, that the two fixed points do not coincide when the exponent characterizing the scale-free (power law) forcing goes to zero. This casts doubts on previous attempts to access the short-range forcing case as a limit of the long-range one within the perturbative RG.

This work is a valuable step for the development of a functional RG approach to turbulence and it provides the framework to improve the results for accessing multi-scaling and for describing the Navier-Stokes turbulence. The manuscript is also clearly presented. I therefore recommend its publication in SciPost Physics.

---

## Editorial Decision

resubmitted